# Numerical Model Generation of Test Frames for Pre-launch Studies of EarthCARE's Retrieval Algorithms and Data Management System

Zhipeng Qu[1], David P. Donovan[2], Howard W. Barker[1], Jason N. S. Cole[1], Mark W. Shephard[1], Vincent Huijnen[2]

[1] Environment and Climate Change Canada, Toronto, ON, Canada
[2] Royal Netherlands Meteorological Institute, De Bilt, Netherlands

*Correspondence to*: Howard W. Barker (howard.barker@canada.ca)

**Abstract.** The *Earth Cloud, Aerosol and Radiation Explorer* (EarthCARE) satellite consists of active and passive sensors whose observations will be acted on by an array of retrieval algorithms. EarthCARE's retrieval algorithms have undergone pre-launch verifications within a virtual observing system that consists of 3D atmosphere-surface data produced by the Global Environmental Multi-scale (GEM) Numerical Weather Prediction (NWP) model, and instrument simulators that when applied to NWP data yield synthetic observations for EarthCARE's four sensors. Retrieval algorithms operate on the synthetic observations and their estimates go into radiative transfer models that produce top-of-atmosphere solar and thermal broadband radiative quantities, which are compared to synthetic broadband measurements thus mimicking EarthCARE's radiative closure assessment. Three high-resolution test frames were simulated; each measures ~6,200 km along-track by 200 km across-track. Horizontal grid-spacing is 250 m and there are 57 atmospheric layers up to 10 mb. The frames span wide ranges of conditions and extend over: i) Greenland to The Caribbean crossing a cold front off Nova Scotia; ii) Nunavut to Baja California crossing over Colorado's Rooky Mountains; and iii) central equatorial Pacific Ocean that includes a mesoscale convective system. This report discusses how the test frames were produced and presents their key geophysical features. All data are publicly available and, owing to their high-resolution, could be used to simulate observations for other measurement systems.

## 1. Introduction

The *Earth Cloud, Aerosol and Radiation Explorer* (EarthCARE) satellite mission, which is scheduled for launch in early- to mid-2024, is a joint venture funded by the European Space Agency (ESA) and Japanese Aerospace Exploration Agency (JAXA) (Illingworth et al. 2015). The combination of Dopplerized cloud profiling radar (CPR), high-spectral-resolution lidar (ATLID), and multi-spectral imager (MSI) will facilitate synergistic retrieves of profiles of cloud, aerosol, and precipitation properties. Broadband top-of-atmosphere (TOA) radiances and fluxes calculated using these profiles will be compared to near-coincidental observations made by EarthCARE's broadband radiometer

(BBR). This radiative closure assessment of retrievals will provide continuous feedback of performance to algorithm developers, as well as guidance to data users.

ESA's pre-launch phase of EarthCARE has relied much on *end-to-end simulation* of measurements, retrievals, and data archiving procedures. The primary objective was to build a virtual observing system in which retrieval algorithms, developed expressly for EarthCARE, get applied to synthetic observations that resemble closely those that will be made by all of EarthCARE's sensors. The initial step of this multi-stage process is definition of atmosphere-surface conditions. The obvious starting point was single homogeneous columns, but this quickly evolved into numerical simulation of realistic conditions for domains that span substantial portions of EarthCARE's planned orbit. These atmosphere-surface conditions are then operated on by instrument simulators that yield synthetic observations suitable for ingestion by retrieval algorithms. One could stop here and assess performance by comparing retrieved geophysical quantities to their simulated counterparts (cf. Mason et al. 2023), but in the real mission this is impossible to do routinely. As such, the next step in the *end-to-end* simulation chain is application of radiative transfer models to retrieved geophysical properties. This produces radiometric quantities that are commensurate with synthetic BBR observations that derive from application of similar radiative transfer models directly to the simulated atmosphere-surface fields. The comparison of these quantities defines the radiative closure assessment of EarthCARE's retrievals.

EarthCARE's data-handling system processes observations into eight "frames" per orbit. As such, frames are ~6,500 km in the along-track direction. Their across-track width is 150 km as defined by the MSI's swath. Almost all measured and retrieved products are reported on the *Joint Standard Grid* (JSG), whose resolutions are ~1 km in both horizontal directions and 0.5 km in the vertical. It was established early on, by EarthCARE's science and engineering teams, that synthetic observations for end-to-end experiments need to cover entire frames and be resolved horizontally to better than 1 km. Environment and Climate Change Canada's (ECCC) Numerical Weather Prediction (NWP) model, known as the Global Environment Multiscale (GEM) model (Côté et al., 1998, Girard et al., 2014), was used to produce three such test frames with horizontal grid-spacings of 250 m and 57 atmospheric layers up to 10 mb. The frames include wide ranges of conditions and extend from: i) Greenland to The Caribbean, crossing a cold front off Nova Scotia; ii) Nunavut to Baja California, crossing over Colorado's Rooky Mountains; and iii) central equatorial Pacific Ocean, including a mesoscale convective system. The primary purposes of this paper is to report on how

these frames were constructed, their cloud, aerosol, and surface properties, as well as adjustments that were made to GEM's initial estimates of ice cloud particle sizes.

60    Full-frame datasets produced by GEM serve as input to the *EarthCARE simulator* (ECSIM) (Voors et al. 2007). ECSIM consists of radiative transfer and instrument models that are coupled to databases of optical and microwave scattering properties. Bulk properties of atmospheric attenuators, such as 3D distributions of GEM's cloud water contents (CWC), are used in conjunction with assumed aerosol/cloud size distributions in order for ECSIM to produce physically-consistent synthetic measurements for each of EarthCARE's sensors. Production of simulated L1

65    EarthCARE data using ECSIM is described by (Donovan et al. 2023). Use of high-resolution full-frame data in ECSIM not only allows assessment of the quality of EarthCARE's retrievals, it also facilitates meaningful estimation of required computational resources and processing times for each algorithm.

The following section discusses how the test frames were defined. This is followed by descriptions of GEM and how it was configured and used for this study. Aerosols and surface optical properties were added to GEM's atmospheres,

70    and these procedures are discussed in sections 4 and 5. Section 6 presents, and to a limited extent verifies, the simulated frames. Section 7 discusses issues with, and subsequent modifications to, GEM's simulated ice clouds. Concluding remarks and information regarding acquisition of test frame data are provided in the final section.

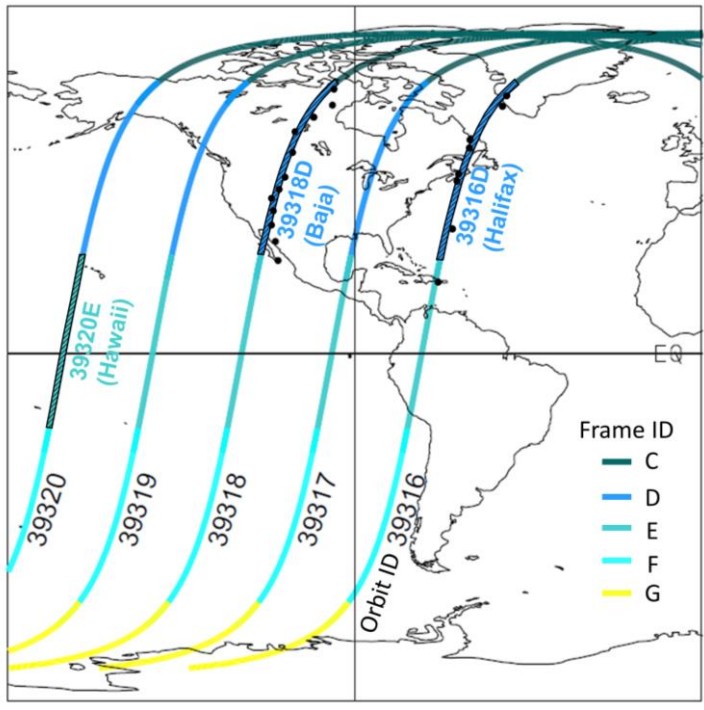

**Figure 1:** Examples of several successively numbered EarthCARE orbits as provided by ESA. Frames are colour-coded with the test frames labelled as 39316D (Halifax), 39318D (Baja), and 39320E (Hawaii).

## 2. Satellite orbit selection

*Figure 1* shows several EarthCARE orbits, numbered 39316 through 39320. An orbit consists of eight frames; each frame's number having an appending letter from A to H which is defined by given ranges of altitude (JAXA, 2017). Frames are colour-coded and measure ~5,000 km along-track and 150 km across-track. All frames selected for testing correspond to local afternoon descending conditions (i.e., opposite to the A-Train). Assuming that night-time atmospheric conditions are not fundamentally different from day-time conditions, night retrievals can be approximated by neglecting MSI solar channels and solar back-ground for ATLID. Test frames should cover wide varieties of clouds, surface, meteorological, and solar illumination conditions. Locations and times needed to initialize simulations of test frames have been established by examining GOES satellite imagery and surface meteorological data. Also, A-Train's active sensor observations had to intersect the frame.

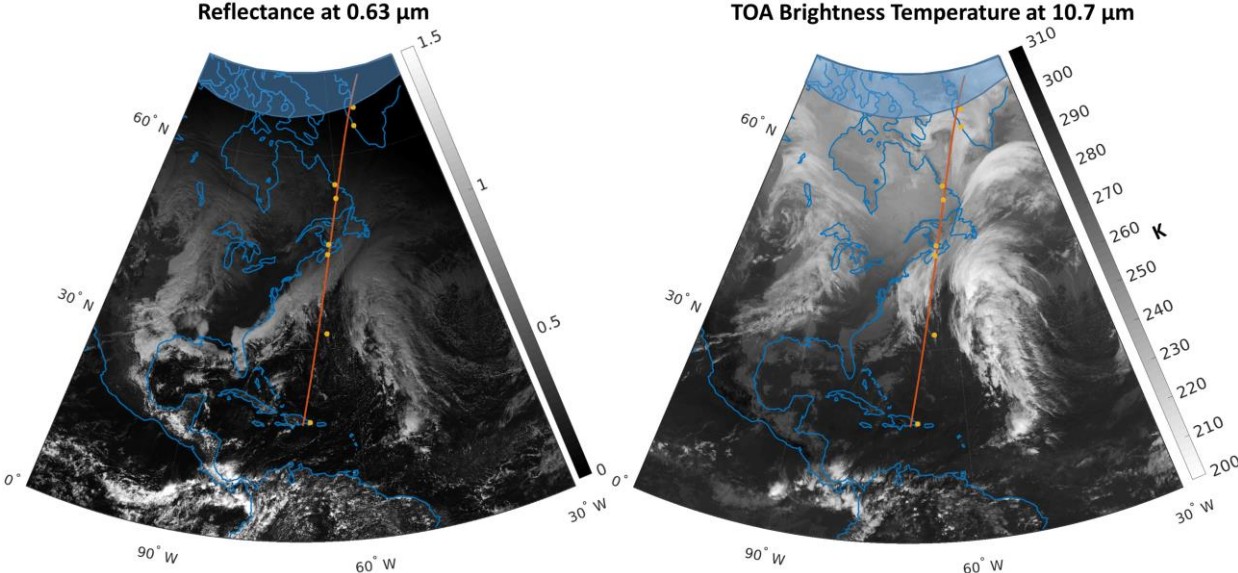

**Figure 2:** GOES-13 TOA reflectance at 0.63 µm and brightness temperature at 10.7 µm on 2014-12-07 at 18h00 UTC. Red lines indicate EarthCARE's track for the *Halifax frame* (see orbit 39316D in *Figure 1*). Yellow dots mark locations of nearby surface meteorological stations. Blue shaded areas indicate GOES-13's northern limit of observation.

**Table 1:** Surface conditions (observed / modelled) near the *Halifax frame* (see orbit 39316D in *Figure 1*) for 2014-12-07 at 18h00 UTC

| station | Temperature (°C) | Dew Point (°C) | Pressure (hPa) | Visibility (km) | Wind Direction | Wind Speed (km/h) | Conditions |
|---|---|---|---|---|---|---|---|
| **Kangerlussaq** | -15.0 / -16.3 | -20.0 / -18.9 | 1006 / 1006 | - | E | 18.5 | Mostly cloudy |
| **Nuuk** | -4.0 / -2.6 | -13.0 / -6.1 | 1000 / 1000 | - | ESE | 33.3 | Low Drifting Snow + Snow |
| **Hopedale** | -17.6 / -15.6 | -23.9 / -21.1 | 1017 / 1022 | - | SW | 31.0 | - |
| **Goose Bay** | -16.0 / -15.5 | -24.0 / -18.6 | 1021 / 1024 | 24.1 | WNW | 22.2 | Scattered Clouds |
| **Charlottetown** | -6.0 / -4.3 | -11.0 / -8.8 | 1029 / 1029 | 24.1 | NW | 27.8 | Mostly Cloudy |
| **Halifax** | -3.0 / -2.7 | -6.0 / -7.2 | 1026 / 1028 | 24.1 | NNW | 31.5 | Overcast |
| **Bermuda** | 24.0 / 22.8 | 18.0 / 18.5 | 1009 / 1010 | - | N | 24.1 | Scattered Clouds |
| **Punta Cana** | 29.0 / 26.6 | 22.0 / 20.7 | 1012 / 1014 | - | NE | 18.5 | Scattered Clouds |

As *Figure 1* shows, frame 39316D extends from southern Greenland, across extreme eastern Canada, and ends in the Atlantic Ocean roughly 500 km north of Dominican Republic. Because it passes close to the city of Halifax, Nova Scotia, it is referred to hereinafter as the *Halifax frame*. *Figure 2* shows GOES-13 reflectances and TOA brightness temperatures for its 0.63 µm and 10.7 µm channels for 2014-12-07 at 18h00 UTC. *Table 1* lists surface conditions reported at 18h00 UTC by several meteorological stations close to the ground-track (see dots on both *Figure 1* and

*Figure 2*). This frame includes no Sun over Greenland, cold surface air over eastern Canada, a cold-front with deep clouds just off the coast of Nova Scotia, and scattered shallow clouds between Bermuda and Dominican Republic.

The second frame, 39318D, is referred to as the *Baja frame*. It stretches from the Canadian Arctic Archipelago, over central North America's Great Plains and Rocky Mountains, and ends near Baja California Sur. *Figure 3* shows GOES-15 imagery and EarthCARE's ground-track for 2015-04-02 at 21h00 UTC. *Table 2* lists surface conditions reported at 21h00 UTC. At the north end of this frame surface conditions were cold with blowing snow and largely cloudless. Through the Canadian Prairies there were low scattered clouds over snow-covered surfaces, while over the

Rocky Mountains skies were very cloudy. In the southern reaches, skies were clear with some cirrus, and surface conditions were warm and very dry.

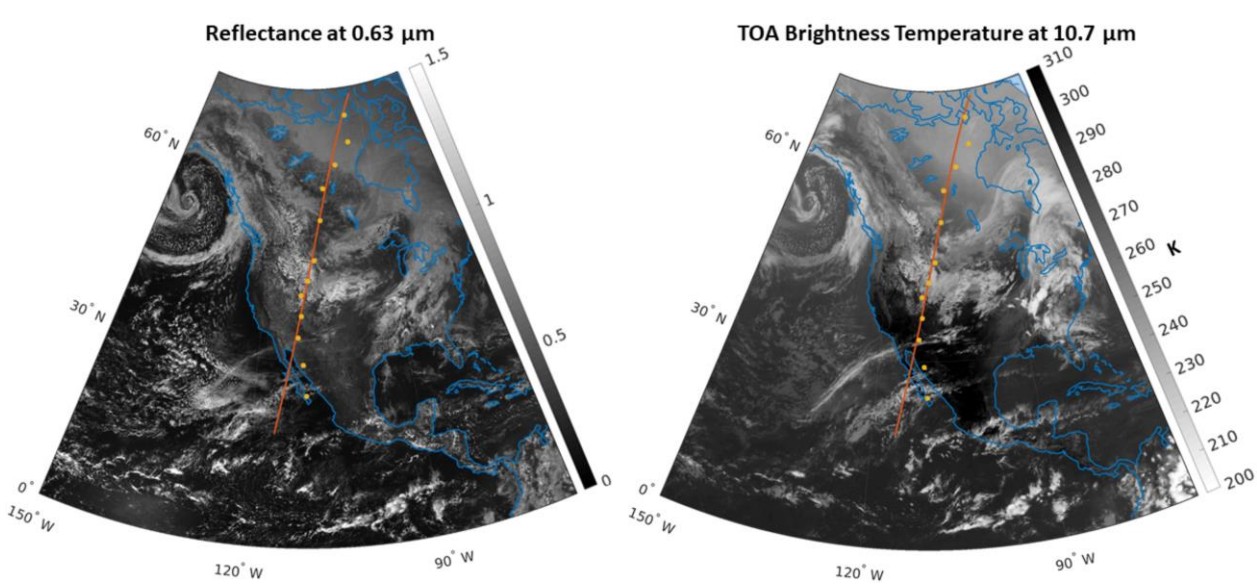

**Figure 3:** As in *Figure 1* but this is GOES-15 imagery for 2015-04-02 at 21h00 UTC. Red lines indicate the *Baja*
*frame* (see orbit 39318D in *Figure 1*).

**Table 2:** As in *Table 1* but these are for the *Baja frame* (see orbit 39318D in *Figure 1*) for 2015-04-02 at 21h00 UTC

| station | Temperature (°C) | Dew Point (°C) | Pressure (hPa) | Visibility (km) | Wind Direction | Wind Speed (km/h) | Conditions |
|---|---|---|---|---|---|---|---|
| Gjoa Haven | -26.0 / -24.6 | -29.0 / -27.9 | 1024 / 1025 | 24.1 | NNW | 18.5 | Ice Crystals |
| Baker Lake | -27.0 / -25.3 | -31.0 / -29.0 | 1018 / 1022 | 4.8 | N | 40.7 | SnowBlowing + Snow |
| Ennadai | -28.2 / -27.2 | -32.2 / -31.6 | 1025 / 1028 | - | NNW | 50 | Blowing |
| Key Lake | -11.0 / -12.1 | -22.0 / -18.3 | 1025 / 1026 | 14.5 | N | 13 | Clear |
| Saskatoon | 0.0 / 1.0 | -10.0 / -4.6 | 1024 / 1022 | 24.1 | NE | 11.1 | Mostly Cloudy |
| Billings | 7.2 / 8.2 | -12.8 / -13.4 | 1021 / 1020 | 16.1 | NW | 25.9 | Scattered Clouds |
| Big Piney | 1.7 / 1.3 | -16.1 / -11.1 | 1017 / 1022 | 16.1 | NW | 13 | Overcast |
| Provo | 8.0 / 0.0 | -6.0 / -18.6 | 1019 / 1024 | 24.1 | WNW | 18.5 | Mostly Cloudy |
| Page | 18.9 / 18.6 | -15.6 / -7.7 | 1008 / 1028 | 16.1 | W | 35.2 | Clear |
| Phoenix | 28.9 / 28.2 | 0.6 / 1.8 | 1010 / 1014 | 16.1 | WNW | 22.2 | Mostly Cloudy |
| Hermosillo | 34.0 / 30.5 | 2.0 / 3.3 | 1012 / 1011 | 16.1 | SSW | 29.6 | Scattered Clouds |
| La Paz | 31.0 / 21.6 | 9.0 / 15.9 | 1013 / 1014 | 16.1 | W | 11.1 | Mostly Cloudy |

The third frame, 39320E, as shown in *Figure 1* and *Figure 4*, crosses the central Pacific Ocean, near Hawaii, on 2015-06-24. It is referred to as the *Hawaii frame*. GOES-15 imagery at 00h00 UTC on 2015-06-24 indicates that the central portion of the frame bisected a mesoscale convective system (MCS). North and south of the MCS, skies were

mostly cloudless with some broken cloud at variable altitudes. There was also a weak frontal system at its southern extremity.

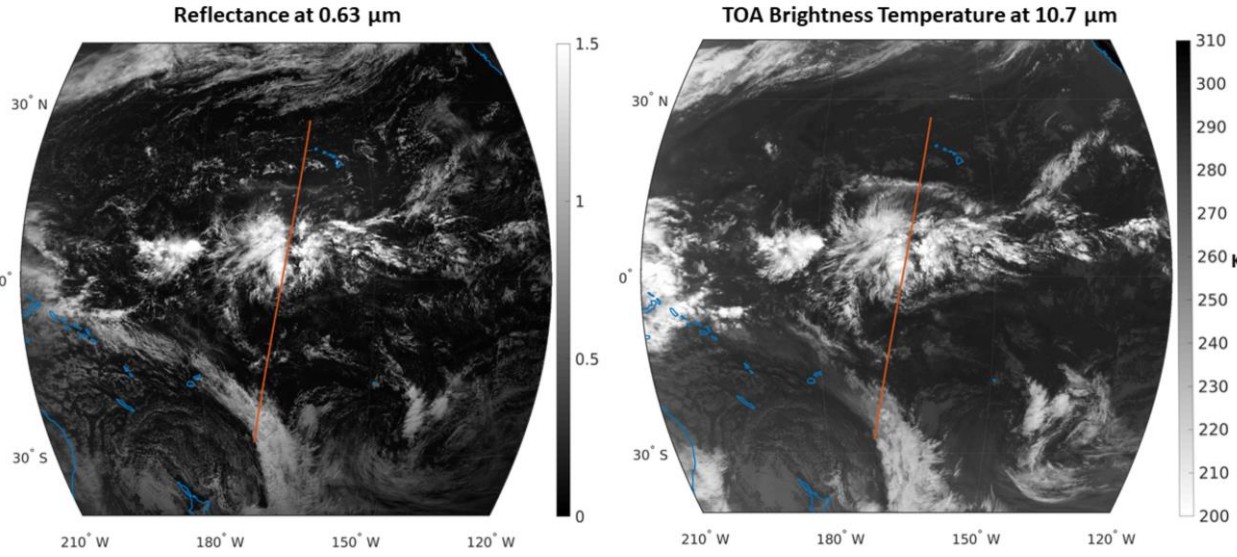

**Figure 4:** As in *Figure 1* but this is GOES-15 imagery for 2015-06-24 at 0h00 UTC. Red lines indicate the *Hawaii*
*frame* (see orbit 39320E in *Figure 1*).

## 3. NWP model set-up

The NWP model used to produce EarthCARE's test frames was ECCC's GEM model (Côté et al., 1998; Girard et al., 2014). GEM's dynamics are formulated in terms of the non-hydrostatic extension of the primitive equations with a terrain-following hybrid vertical grid. It can be run as a global model or a limited-area model and is capable of one-

way self-nesting. For this work, GEM ran with four nested domains at horizontal grid-spacings $\Delta x$ of 10, 2.5, 1, and 0.25 km, with 79 hybrid levels for the 10 km outer-domain and 57 for the other three. The global analysis data used in ECCC's Global Deterministic Prediction System (GDPS) (Buehner et al. 2015) were used as the initial condition for the outermost simulation domain at 10 km horizontal grid-spacing. The GDPS predictions are also used as the lateral boundary conditions with the nesting method described in Thomas et al. (1998).

The simulations at $\Delta x$ of 2.5, 1 and 0.25 km used Milbrandt and Yau's (2005a,b) double-moment bulk cloud micro-physics scheme (referred to hereinafter as MY2), which predicts mass and number mixing ratio for each of six hydrometeors classes: non-precipitating liquid droplets; ice crystals; rain; snow; graupel; and hail. For the $\Delta x = 10$ km domain, the Kain–Fritsch (KF) deep convection scheme (Kain and Fritsch 1990; 1993) was used. Its liquid and ice CWCs are passed later to MY2 as non-precipitating liquid droplet and ice crystal categories.

In addition to MY2 and KF, a planetary boundary-layer scheme can also produce liquid and ice clouds along with fractional cloudiness for cumulus and stratocumulus clouds (Bélair et al. 2005). Moreover, a shallow convection scheme (Bélair et al. 2005) also supplies estimates of liquid and ice CWCs and cloud fractions for cells with shallow cumulus. Both schemes are used in all domains.

The atmospheric turbulence is parameterized with a turbulent kinetic energy (TKE) scheme (Benoit et al. 1989,

Bélair et al. 2005) named MoisTKE. For the simulations with 250 m horizontal grid-spacing, a modified mixing length with an asymptotic value based on the horizontal grid size [$\lambda_0 = 0.23(\Delta x \, \Delta y)^{\frac{1}{2}}$] is used. The readers are refer to Leroyer et al. (2014) for more details.

It was simplest to align GEM's *computational equator* approximately along EarthCARE's orbit, and divide 6,200 km long frames into 13 non-overlapping inner-most domains ($\Delta x = 0.25$ km) and run them separately: 11 segments at

500 km along-track and both end segments at 350 km (all are 200 km wide). The downscaling transitional domains at $\Delta x$ of 2.5 km and 1 km adapt themselves to the locations of the $\Delta x = 0.25$ km domains (both domains at $\Delta x$ of

2.5 km and 1 km are repeated 13 times). A common $\Delta x = 10$ km domain was used for all 13 segments. *Figure 5* illustrates this configuration, and *Table 3* summarizes domain sizes and $\Delta x$. Finally, the 13 inner-most domains are simply concatenated to form 6,200 km frames. While this forms discontinuities, they are not a serious hindrance for the task at hand.

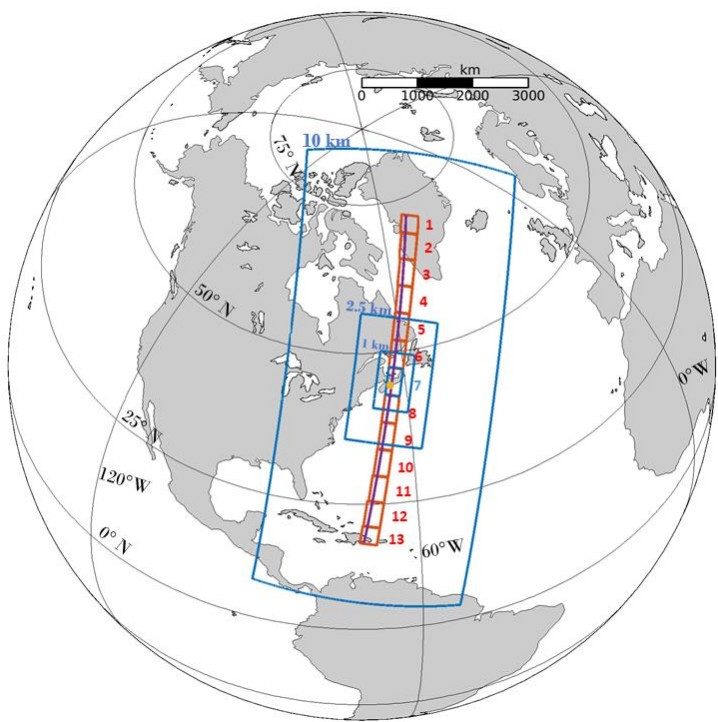

**Figure 5:** Downscaling domains for the *Halifax frame* (39316D). Blue rectangles delineate successive downscaling domains that culminate in the 7[th] innermost domain whose horizontal grid-spacing is $\Delta x = 0.25$ km. Red rectangles are the 12 other innermost domains for this frame. EarthCARE's ground-track is indicated by the purple line, which is 60 km west of centre.

Simulations for the *Halifax frame* (39316D) were initialized at 12h00 UTC on 2014-12-07 and saved at 17h30 UTC. Likewise, the *Baja frame* (39318D) simulations were initialized at 12h00 UTC 2015-04-02 and saved at 21h00 UTC, while the *Hawaii frame* was initialized at 12:00 UTC 2015-06-23 and saved at 00h00 UTC on 2015-06-24. Data for all three frames are publicly available. Variables include CWC, number concentration, and effective radius $R_{eff}$ for the six aforementioned hydrometeor types. Saved variables are listed in the Appendix.

**Table 3:** Sizes of GEM's downscaling domains (see *Figure 5*) and their horizontal resolutions

| description | domain size (km) along-track x across-track | number of vertical layers | horizontal grid-spacing (km) |
|---|---|---|---|
| downscaling domain 1 | 8,600 x 3,600 | 79 | 10 |
| downscaling domain 2 | 2,250 x 1350 | 57 | 2.5 |
| downscaling domain 3 | 1000 x 600 | 57 | 1 |
| innermost domains 1 and 13 | 350 x 200 | 57 | 0.25 |
| innermost domains 2 to 12 | 500 x 200 | 57 | 0.25 |

## 4. Shortwave optical properties for land surfaces

As the additional data for pre-launch studies of EarthCARE, GEM's snow-free surface albedos were replaced by
those based on MODIS's MCD43GF 1 km resolution bidirectional reflectance distribution function (BRDF) product

for the period 2002 to 2013 (Schaaf et al. 2002). These data were interpolated, via nearest neighbour, to 0.25 km.

Conditions for the *Baja* frame are shown here because it is primarily over land; the others are mostly over ocean.

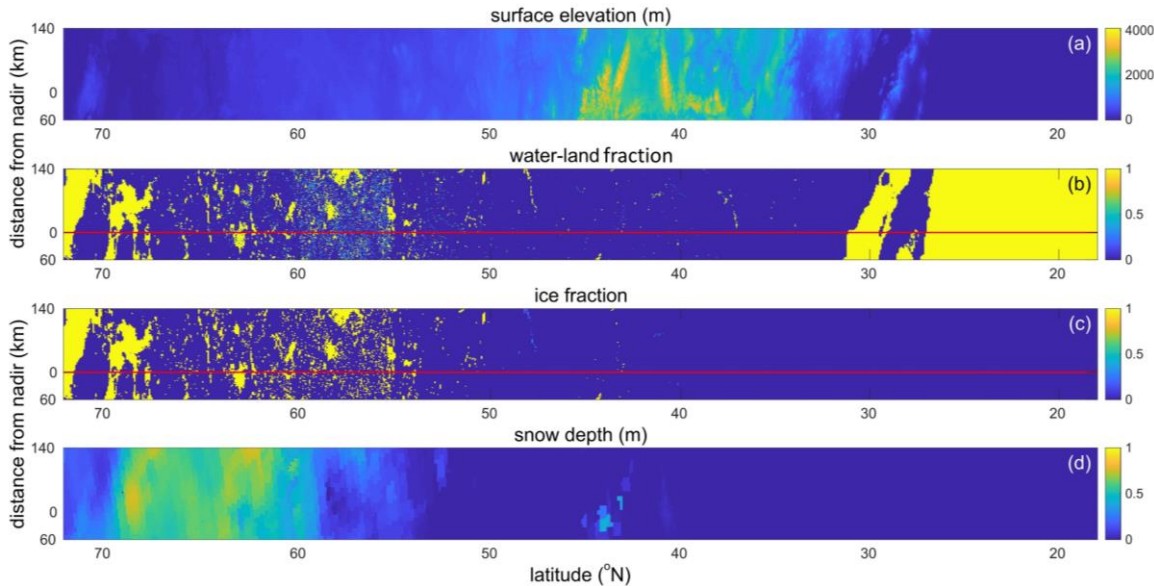

**Figure 6:** All panels are for the *Baja frame* (see *Figure 1* and *Figure 3*) and each panel's title is self explanatory. For
(b) and (c), blue (fraction of 0) corresponds to entirely land and yellow (fraction of 1) to either entirely water or
entirely ice.

*Figure 6*a illustrates the wide range of surface conditions for the *Baja frame*. The Rooky Mountains are crossed near

$40° N$, but this being mid-springtime only small amounts of mountain snow remain. *Figure 6b* highlights the distri-

bution of freshwater lakes in the Canadian Shield; *Figure 6c* shows that most are frozen and snow-covered. From

*Figure 6c* it is clear that shallow snow covers most of the Canadian Prairies with deeper snow north of the tree-line, which for this frame is close to 60° N (see 0.63 μm reflectances in *Figure 3*).

Following Schaaf et al. (2002), spectral-dependent black-sky albedos are defined as

$$
\begin{aligned}
\alpha_{bs}(\theta_0) = \alpha_1 &+ \alpha_2\left(-0.007574 - 0.070987\theta_0^2 + 0.307588\theta_0^3\right) \\
&+ \alpha_3\left(-1.284909 - 0.166314\theta_0^2 + 0.041840\theta_0^3\right),
\end{aligned}
\tag{1}
$$

where $\theta_0$ is solar zenith angle in radians, and $\alpha_1$, $\alpha_2$, and $\alpha_3$ are separate sets of spectral-dependent BRDF kernel

weights for spectral ranges 300 - 700 nm and 700 - 50,000 nm. Corresponding white-sky albedos are defined as

$$
\alpha_{ws} = \alpha_1 + 0.189184\alpha_2 - 1.377622\alpha_3.
\tag{2}
$$

*Figure 7* shows $\alpha_{ws}$ for the *Baja frame*. Note that because these are snow-free values, they tend to be largest in the southern areas; especially in the near-IR for forests of western Colorado and deserts of Arizona and Sonora. With both variable surface elevation and surface albedos, this frame represents a stringent test for retrieval algorithms, as

opposed to the more straightforward ocean surfaces that dominate the other two frames.

For snow/ice covered areas, snow depth and ice fraction and/or surface land type should be used to determine albedo of snow/ice covered surfaces. For ocean and lakes, wind speed is used to determine surface albedo. To determine surface emissivity for different types of surface, Huang et al.'s (2016) surface emissivity climatology was used. Surface albedo and emissivity are discussed further in another paper on broadband radiative quantities (ACM-COM

and ACM-RT products) in this special issue (Cole et al. 2023).

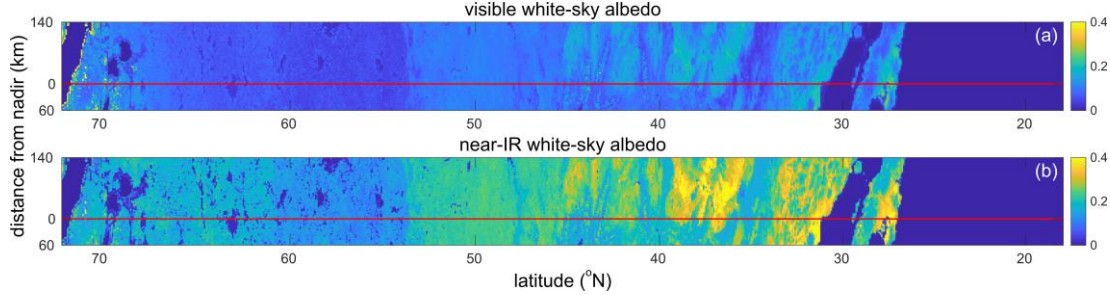

**Figure 7:** (a) Visible white-sky albedos, as defined in (2), for snow-free land for the *Baja frame*. (b) Same as (a) except these are near-IR values. EarthCARE's nadir-track is shown by red lines.

**5. Aerosol properties**

The ECSIM scene creation process requires 3D distributions of aerosol size distributions. As GEM lacks interactive aerosol tracers and chemistry, aerosol fields were added to the test scenes using information from the Copernicus Atmosphere Monitoring Service (CAMS) (Flemming et al. 2017). The CAMS data was at a resolution of 0.5 by 0.5 latitude-longitude degrees and 60 hybrid sigma model levels. The aerosol scheme implemented within ECSIM follows the *Hybrid End-To-End Aerosol Classification* (HETEAC) approach of defining a certain set of basic aerosol types with associated e.g. size distributions, refractive indices and optical properties that, when weighted and summed, yield adequate representations of a wide range of observed aerosol optical properties (Wandinger et al. 2016; Wandinger et al. 2023). *Table 4* lists the CAMS aerosol fields, and the *Supplementary Material* section provides a detailed description of the mapping between CAMS fields and ECSIM/HETEAC scattering types. It also provides more details regarding aerosol representation.

**Table 4:** Aerosol classes from the Copernicus Atmosphere Monitoring Service (CAMS)

| aerosol class | description |
|---|---|
| DD1-DD3 | Dust (in different size intervals) |
| SS1-SS3 | Seat Salt (in different size intervals) |
| SO4 | Sulphate aerosol |
| BCB | Fine mode strongly absorbing aerosol |
| OMB | Weakly absorbing aerosol |

**6. Results: GEM simulations and verification**

The purpose of this section is to show selected results that characterize the EarthCARE test frames produced directly by GEM. Post-simulation adjustments were made to ice microphysical properties as described and shown in both section 7 and *Supplementary Material*.

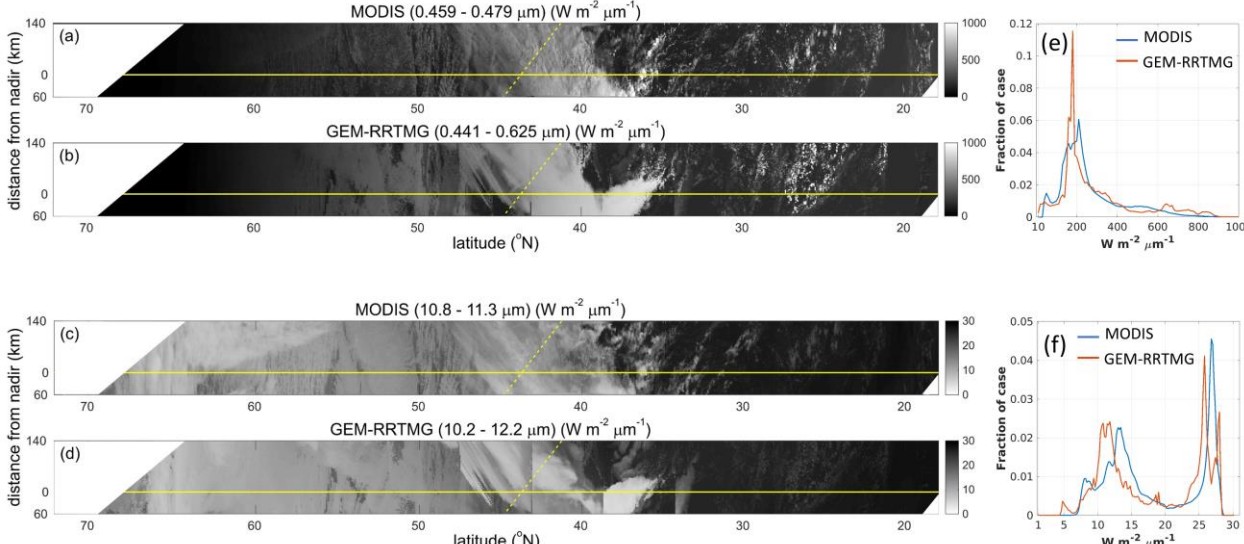


**Figure 8:** (a) MODIS TOA flux (approximated by radiance*π) for band 3 (459 - 479 nm) between 17h15 and 17h35 UTC on 2014-12-07. (b) RRTMG simulated upward TOA flux for 441.5 - 625 nm for GEM's simulation of the *Halifax frame*. (c) as in (a) but for band 31 (10.8-11.3 μm). (d) in as but for wavelengths 10.2 - 12.2 μm. Solid and dashed yellow lines indicate EarthCARE's and CloudSat's nadir-tracks. Blank areas are outside MODIS's field-of-view. (e) frequency distributions of fluxes for band 3 (bin size of 10 W m$^{-2}$ μm$^{-1}$). (f) as in (e) but this is for band 31 (bin size of 0.2 W m$^{-2}$ μm$^{-1}$).

### 6.1. Halifax frame

*Figure 8a* and *c* show MODIS spectral fluxes (MYD02HKM product; MCST 2017a) for 0.459 - 0.479 μm and 10.8 - 11.3 μm for the *Halifax frame*. Key cloud-related features are a cold front between 40°N and 45°N, scattered clouds to its south, and mostly overcast conditions to its north. *Figure 8b* and *d* show TOA spectral fluxes for two wavebands, close to MODIS's bands, as simulated by the Rapid Radiative Transfer Model for GCMs (RRTMG - Mlawer et al. 1997; Iacono et al. 2000; 2008) using GEM data. At large-scales, GEM did well with respect to cloud occurrence. *Figure 8e* and *f* show distributions of visible and infrared spectral fluxes, respectively. While the distributions of fluxes derived from observations and models follow similar patterns, there are some notable differences in the imagery. For the GEM scenes, discontinuities, stemming from the stitching together of the semi-independent high-resolution inner-most domains, are clearly visible across the frontal system. They do not pose a serious problem for the task at hand.

Near 38°N GEM's longwave fluxes are significantly less than MODIS's. This is because GEM simulated widespread convection in this area whereas MODIS only observed isolated convective cells. This is also evident in *Figure 8e* and

*f* as GEM shows higher frequencies around 800 W m$^{-2}$ μm$^{-1}$ and 5 W m$^{-2}$ μm$^{-1}$, respectively. This is also apparent in *Figure 9*, which shows cloudtop altitudes both inferred from MODIS radiances (Platnick et al. 2015) and computed by the MODIS simulator of the Cloud Feedback Model Intercomparison Project (CFMIP) Observation Simulator Package (Bodas-Salcedo et al. 2011; abbreviated as the *COSP simulator*).

GEM's cloudtop altitudes are too high for low clouds between latitude 20°N and 30°N and 50°N and 55°N; most are
near 750 hPa, whereas MODIS's values are mostly near 920 hPa. This can also be inferred from *Figure 8f* in which higher frequencies of infrared spectral fluxes from GEM are found between 23 and 25.5 W m$^{-2}$ μm$^{-1}$ for the southern section and between 10 and 12 W m$^{-2}$ μm$^{-1}$ for the northern section. Additionally, *Figure 9c* shows that GEM underestimates the amount of mid-level clouds between 500 and 600 hPa in the region between latitude 55°N and 62°N.

These mentioned above, differences cannot be explained by the slight time difference between MODIS observation
(~17h20 UTC) and GEM simulation time (17h30 UTC). On the other hand, it is common for NWP models to simulate some characteristics of cloud systems quite well yet show temporal/spatial displacements relative to observations (e.g., Qu et al. 2018). The goal when simulating these test scenes was, however, to produce large, well-resolved tracts of realistic clouds; emphasis on exactly what happened was secondary.

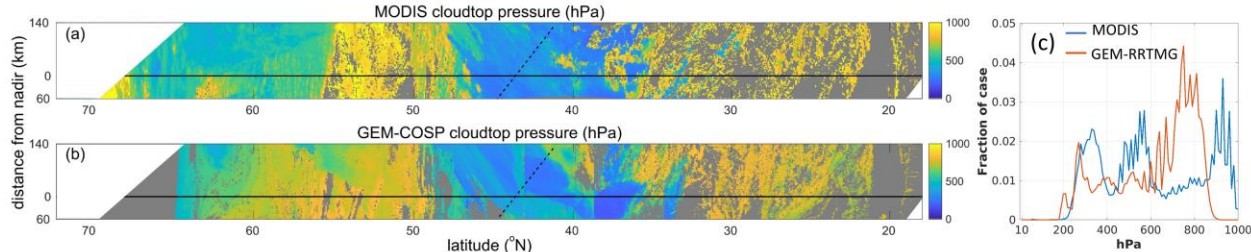

**Figure 9:** (a) MODIS cloudtop pressure (MYD06_L2 product) between 17h15 and 17h35 UTC on 2014-12-07. Blank areas are outside MODIS's field-of-view. (b) GEM's cloudtop pressure for the *Halifax frame* based on COSP's MODIS simulator. Grey area in the northern portion has $\theta_0 > 90°$ and so no COSP values. (c) frequency distributions of cloudtop pressure (bin size of 10 hPa).

*Figure 10*b and c show cross-sections of ice CWC inferred from CloudSat radar reflectivities and simulated by GEM,
respectively (for the transect indicated in *Figure 8*). While the cross-sections intersect only at latitude 43.6°N, the general forms of the fields agree well, and not just in the immediate vicinity of the intersection point. Between 42°N and 43°N, GEM produces a large amount of solid precipitation whereas in CloudSat data, due to the ground clutter, there is no reliable retrieval available. Unfortunately, CloudSat's retrieval of liquid CWC is problematic (e.g., Li et

al. 2018) and are not used here to assess GEM's. *Figure 10*a shows the ice water path (IWP) vertical integrals of

values in *Figure 10*b and c. At and around the intersection point, CloudSat's IWP values are much larger than

GEM's, but again, the forms of their curves are fairly similar. The edge of GEM's inner-domain near 43°N is abun-

dantly clear.

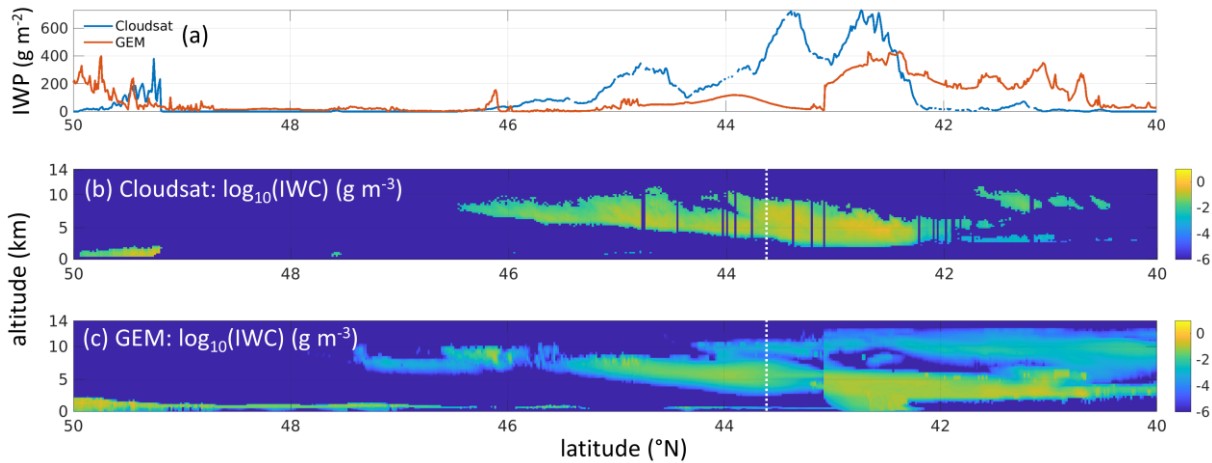

**Figure 10:** (a) Ice water path (IWP) inferred from CloudSat observations at 17h21 UTC on 2014-12-07 as it crossed the *Halifax frame* between latitudes 41°N - 44°N (dashed yellow line in *Figure 8*), and as simulated by GEM along the nadir-track (solid yellow lines in *Figure 8*). (b) and (c) are ice CWC for CloudSat and GEM, respectively. Dashed white line indicates where CloudSat's and EarthCARE's tracks intersected.

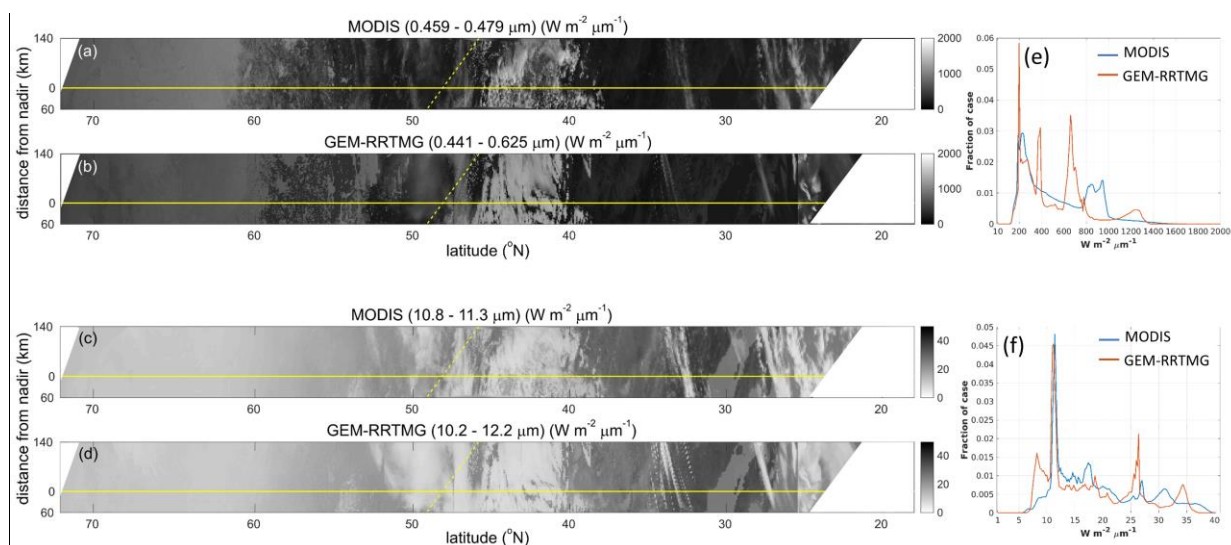

**Figure 11:** As in *Figure 8* except these are for the *Baja frame*. MODIS observations were between 20h10 and 20h30 UTC on 2015-04-02.

### 6.2. Baja frame

*Figure 11* compares MODIS TOA fluxes to those computed by RRTMG acting on GEM data for the *Baja frame*. As with the *Halifax frame*, agreement is generally good, though GEM's fields exhibit some peculiarities. For instance, GEM's fluxes associated with clouds are less variable than MODIS's; especially between 40°N and 50°N. This could
be due to both GEM's clouds being simply too homogeneous due to missing mesoscale forcing (Stensrud and Gao, 2010) or RRTMG's use of 1D radiative transfer models (Barker et al. 2017). Also, the thin high clouds near latitude 32°N, which are also evident in *Figure 12* and positioned well in space, show an on-off pattern that is not seen in the observations. Furthermore, near latitude 55°N GEM failed to produce the very thin, but extensive, clouds below 800 hPa. This is most apparent in *Figure 12*. GEM's overestimation of cloudtops close to 400 hPa near latitude 50°N is
consistent with *Figure 12c* and *Figure 11e* and *f* which show significant overestimations of fluxes between 620 and 730 W m$^{-2}$ μm$^{-1}$ for visible band and between 7 and 10 W m$^{-2}$ μm$^{-1}$ for infrared band. Note too, that the discontinuities that stem from stitching together GEM's innermost domains are less apparent for this frame than they are for the *Halifax frame*, though the discontinuity near 26°N is notably bad for it stands out in both visible and IR imagery.

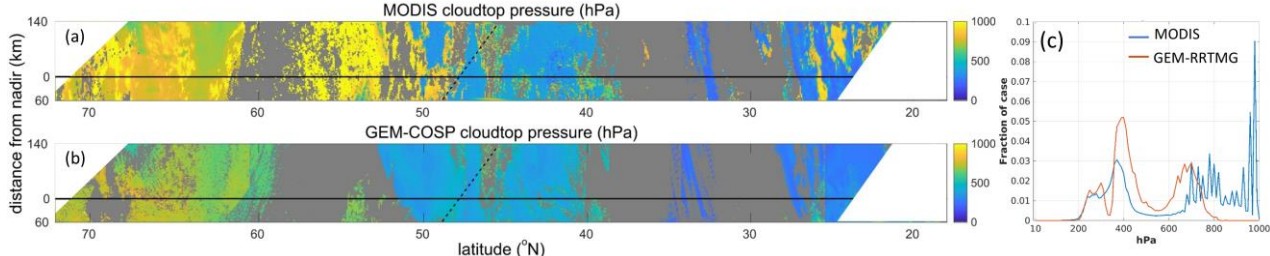


**Figure 12:** As in *Figure 9* except these are for the *Baja frame*. MODIS cloudtop pressure are retrieved between 20h10 and 20h30 UTC on 2015-04-02.

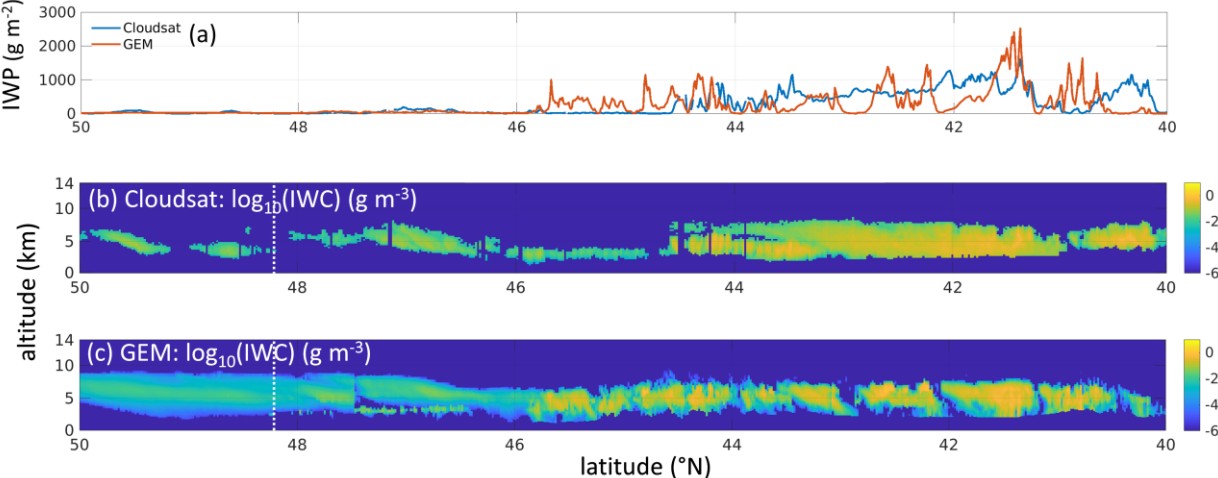

**Figure 13:** As in *Figure 10* except these are for the *Baja frame*. CloudSat's track indicated by the dashed yellow line in *Figure 11* is between latitudes 45°N - 49°N.

Despite these discrepancies, *Figure 13* shows that in the vicinity of where the satellite tracks intersect, vertical realizations of clouds from both GEM simulations and CloudSat retrievals indicate smooth mid-level low density clouds, although GEM's are more extensive. The altitudes of GEM's clouds over the Rooky Mountains are also in fair agreement with CloudSat's. Unlike the *Halifax frame*, the magnitudes of modelled and "observed" IWPs agree quite nicely, in general.

### 6.3. Hawaii frame

*Figure 14* shows that for the *Hawaii frame*, GEM's positionings and approximate intensities of cloud systems near the Equator and ~25°S agree well with the MODIS observations. The harsh discontinuity in GEM's string of inner-most domains near 2°S is due to a lack of high ice cloud, as seen in *Figure 15*, which likely stems from the lack of information, in the form of reduced outflow of high cirrus, coming into the sub-domain from the equatorial mesoscale system. Likewise, near 15°N the lack of upper-level cloud in GEM could be because this sub-domain was too disconnected from the mesoscale system to the south. The lack of high cloud in the simulation can be inferred from *Figure 14*e which shows an overabundance of fluxes by GEM near 200 W m$^{-2}$ μm$^{-1}$; a value that resembles TOA visible fluxes from ocean surface. This is also seen in *Figure 14f* and *Figure 15*c. Again, however, the point of this section is to show the gross verisimilitude of the test frames and hence their suitability for EarthCARE algorithm assessments.

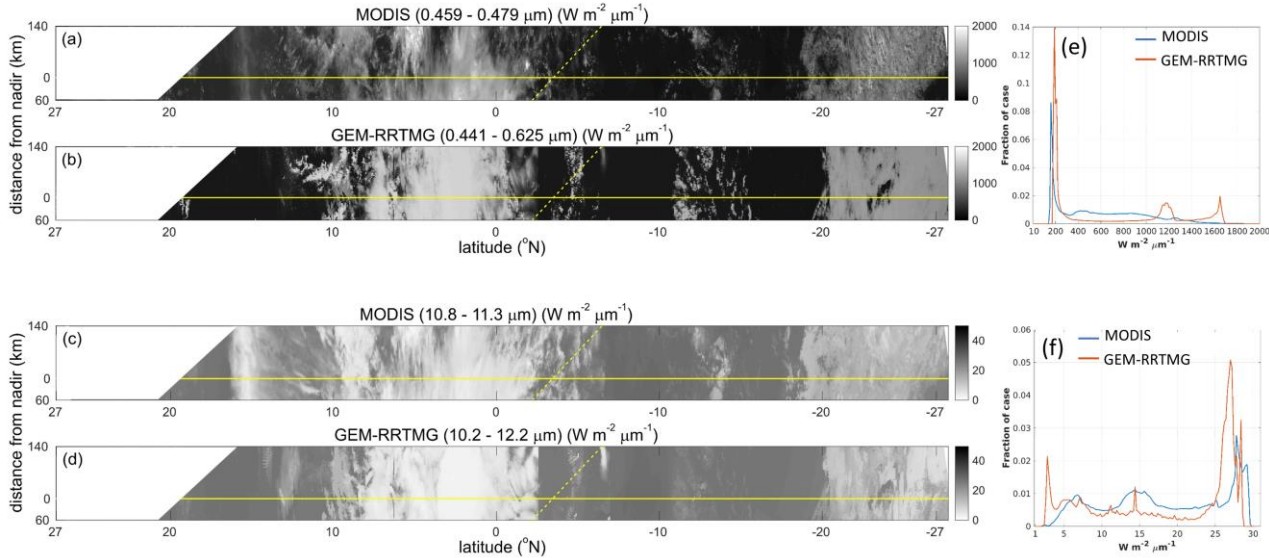

**Figure 14:** As in *Figure 8* except these are for the *Hawaii frame*. MODIS observations were between 00h35 and 00h55 UTC on 2015-06-24

As *Figure 16* shows, despite CloudSat's track intersecting EarthCARE's well south of the mesoscale system situated near the centre of the *Hawaii frame*, the system was sufficiently large in the zonal direction that CloudSat's sampling of it can be compared to EarthCARE's sampling of GEM's simulation. The regions of high ice CWC values for the two samples match extremely well both vertically and horizontally. The distribution of ice CWC inferred from CloudSat reflectivities is very narrow while GEM's is much broader with many extremely small values ($10^{-4}$ to $10^{-5}$ g

$m^{-3}$) that are below the detection threshold of COSP. Aside from the huge spike in IWP for GEM near 3°N, which obviously included some precipitation, the magnitude and forms of the curves for CloudSat and GEM agree well.

What might appear to be a deficiency with GEM is the extreme lack of texture in the visible reflectance of cloud associated with the frontal system in the south of frame. As with the other frames, however, it is entirely likely that the smoothness GEM's field stems from application of a 1D radiative transfer model (see Barker et al. 2017). This is

addressed explicitly in other papers in this special issue (Cole et al. 2023).

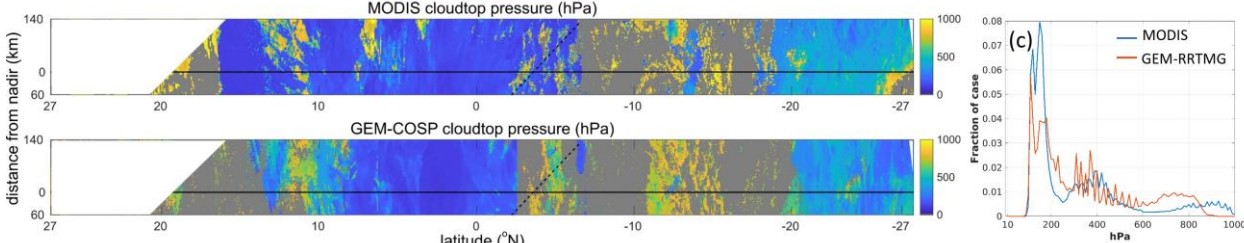

**Figure 15:** As in *Figure 9* except these are for the *Hawaii frame*. MODIS cloudtop pressures were retrieved from observations between 00h35 and 00h55 UTC on 2015-06-24

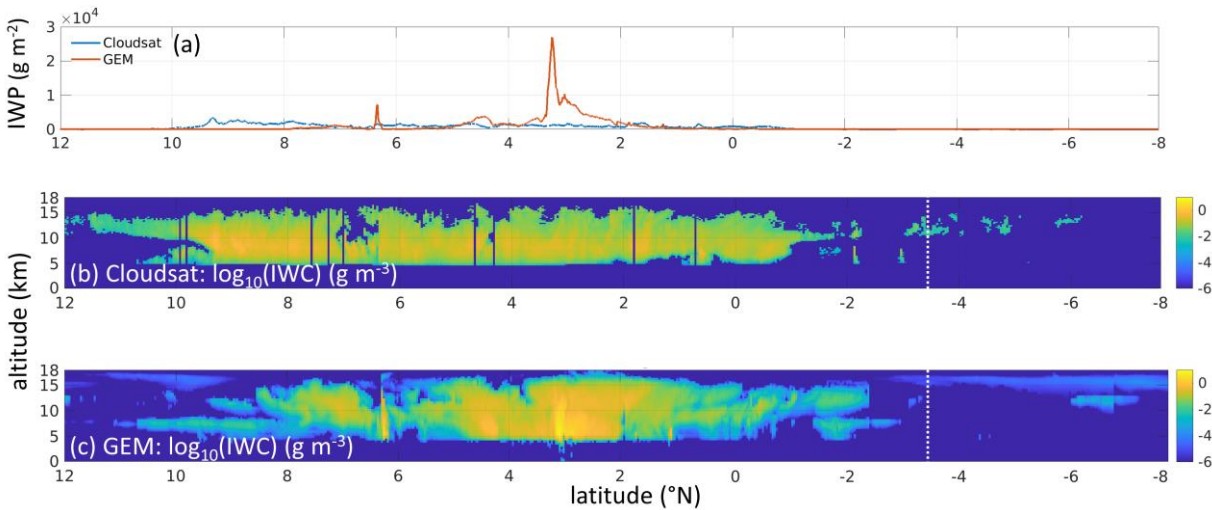

**Figure 16:** As in *Figure 10* except these are for the *Hawaii frame*. CloudSat's track indicated by the dashed yellow line in *Figure 14* is between latitudes 1.5°S - 6.5°S.

## 7. Alterations of GEM's ice crystal sizes

As GEM's scenes are to be input to ECSIM (Voors et al. 2007) to simulate synthetic L1-level measurements for ATLID, CPR, MSI, and BBR (Donovan et al. 2023), it is important that not only macrophysical cloud properties be

realistic, for phenomena such as solar RT and CPR nonuniform beam filling (Tanelli et al. 2002), but cloud microphysical properties, such as size distributions and mass-diameter relationships, have to be, too. While the macrophysical cloud properties simulated by GEM were deemed satisfactory in the previous section, it was clear that there were shortcomings with its predicted ice cloud microphysical properties (cf. Qu et al. 2018). These deficiencies had a demonstrably negative impact on the realism of ECSIM's simulated measurements, and so adjustments to ice particle

sizes were needed.

Basically, GEM predicts too many overly small ice crystals with $R_{eff} < 10$ μm. The cause of this appears to be overestimation of ice crystal number concentrations near cloudtops. Currently, secondary ice production (SIP) mechanisms are poorly understood (Field et al., 2017; Korolev and Leisner 2020), and while at least 6 SIP mechanisms are known (Korolev et al. 2020), only the Hallett-Mossop process (Hallett and Mossop 1974, Mossop and Hallett 1974) is parametrized in the MY2 scheme. It appears as though ice number concentrations in GEM's simulations are systematically underestimated near, or just above, the melting layer. Hence, cloud glaciation times will be too long, and an excess of liquid droplets will be sent too high by updrafts. In the current scheme, droplets will eventually be converted into ice crystals via homogenous freezing, and this will produce very high concentrations of small crystals at altitude. While ongoing studies aim to improve representations of SIP (e.g., Huang et al., 2021, Qu et al. 2022a), they are not yet ready for use in GEM. As such, more manual alterations to GEM's ice crystal sizes were needed.

To improve the realism of the synthetic observations, the following adjustments were made to the GEM's original fields:

1. Implicit liquid CWCs (see *Table A1* in the Appendix) at temperatures < 273 K were set to zero thereby reducing unrealistically large amounts of super-cooled droplets;

2. Implicit ice CWCs (see *Table A1* in the Appendix) were removed because crystal $R_{eff}$ were artificially fixed at 15 μm;

3. The mass-dimension relationships used by GEM for ice and snow were replaced by those described in Erfani and Mitchell (2006) and the functional form of ice particle size $D$ distribution was changed. Specifically, it was altered by multiplying by a factor of $D^4$ which had the effect of increasing mean $D$. Following these adjustments, particle number densities were recalculated subject to conservation of GEM's original total ice and snow water contents.

These alterations were found to produce significant, albeit from a qualitative perspective, improvements to GEM's simulated cloud properties. For example, considering the issue of too many, too small ice crystals, *Figure 17* shows

the relationship between cloud and ice particle $R_{eff}$ before and after the adjustments listed above. It can be seen that

the population of small crystals at temperatures above 245 K has been eliminated, whilst below 240 K, minimum

particle sizes after adjustments exceed 10 μm. Moreover, the distribution of $R_{eff}$ after adjustments is more con-

sistent with the phase-space indicated by real observations (e.g., Donovan and van Lammeren 2001; Wyser 1998).

Lines in *Figure 17* are from parametrizations. While many observation-based parametrizations of ice crystal size

distribution exist, they exhibit only moderate agreement, and so cannot be used to fully support the credibility of

adjusted $R_{eff}$ . It can be concluded, however, with some certainty, that the above adjustments removed unrealistically

small ice crystals and that the resulting temperature distribution of $R_{eff}$ is in fair agreement with observations.

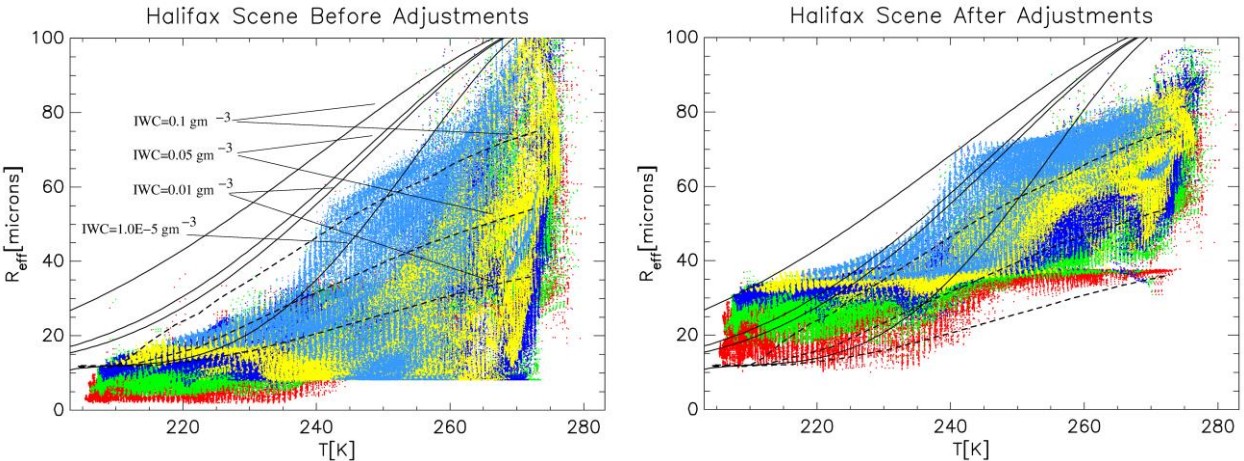

**Figure 17:** Effective radius $R_{eff}$ as functions of temperature for GEM's results both before and after application of
adjustments discussed in the text. $R_{eff}$ is defined in terms of the mass and cross-sectional area of crystals following
Donovan and van Lammeren (2001). Solid lines in the "before" panel correspond to the parametrization described in
Wyser (1998), while dotted-lines follow Donovan and van Lammeren (2002). Colours of dots correspond to different
ranges of ice water content (IWC) (g m$^{-2}$): red → IWC < 0.0001, green → 0.0001 < IWC < 0.001, blue → 0.001 <
370                        IWC < 0.01, yellow → 0.01 < IWC < 0.1, light-blue → 0.1 < IWC.

Censoring the implicit super-cooled liquid and ice water as well as the adjustment to ice and snow $R_{eff}$ have im-

portant consequences for the vertical structure of optical extinction. This can be seen in *Figure 18* where nadir cross-

sections of $R_{eff}$ and extinction at 355 nm (the operating wavelength of ATLID) are shown both before and after

adjustments were performed. Increases in $R_{eff}$ and droplet extinction for clouds poleward of 50°N and at altitudes

below 5 km stem from a combination of removing super-cooled implicit water and increases to ice and snow $R_{eff}$.

The reduction of cloud extinction, especially near cloudtops between 35°N and 45°N is mainly a consequence of

increasing $R_{eff}$ of ice particles.

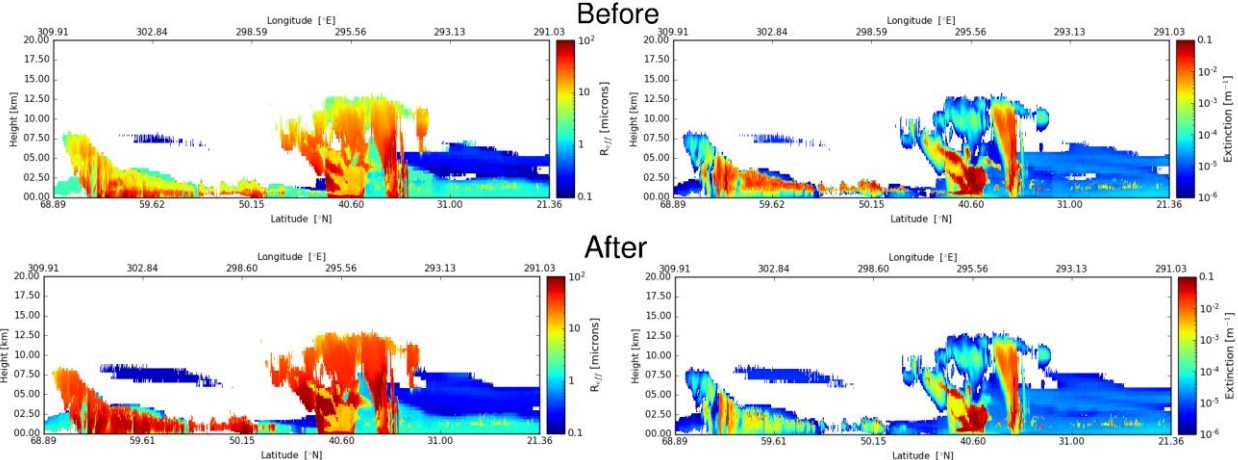

**Figure 18:** Nadir cross-sections of $R_{eff}$ and 355 nm extinction before and after making adjustments described in the
text. Note that the changes in the aerosol regions e.g. the elevated layer north of 50 Degrees at around 7.km are due
to technical updates to the aerosol processing between the "Before" and "After" data not related to the ice-cloud
adjustments.

Impacts of these adjustments can be seen in *Figure 19*, which shows fractions of cases as functions of effective

radius and cloudtop pressure. *Figure 19a* is for MODIS retrievals (MYD06_L2) and shows that for most cases with

cloudtop pressure between 200 and 400 hPa effective radii are between 30 and 50 μm. *Figure 19b* shows that for

COSP simulations based on GEM data, most ice clouds for the same cloudtop pressures have effective radius smaller

than 15 μm. After applying the adjustments, however, COSP values improve significantly with most effective radii

between 30 and 50 μm. Though not shown, similar impacts exist for the Baja and Hawaii scenes.

In addition to these improvements in cloud optical properties, the same adjustments were found to improve the

realism of cloud properties that are relevant to simulation of CPR observations. For example, after applying the

adjustments the relationship between ice CWC and simulated radar reflectivity now falls in phase-space that agrees

well with real observations (e.g., Matrosov and Heymsfield 2017; Heymsfield et al. 2005). *Figure 20* shows the IWC

vs. Ka-band reflectivity for the nadir Halifax scene path. It can be seen that after adjustment (*Figure 20b*) the best-fit

line of GEM data compares well with the relationships shown in Fig 4 of Matrosov and Heymsfield (2017). The

agreement is even more striking when distributions of data shown in Matrosov and Heymsfield (2017) are considered instead of just best-fit lines. Various cross-sections of adjusted GEM+CAMS-derived fields can be found in the *Supplementary Material*.

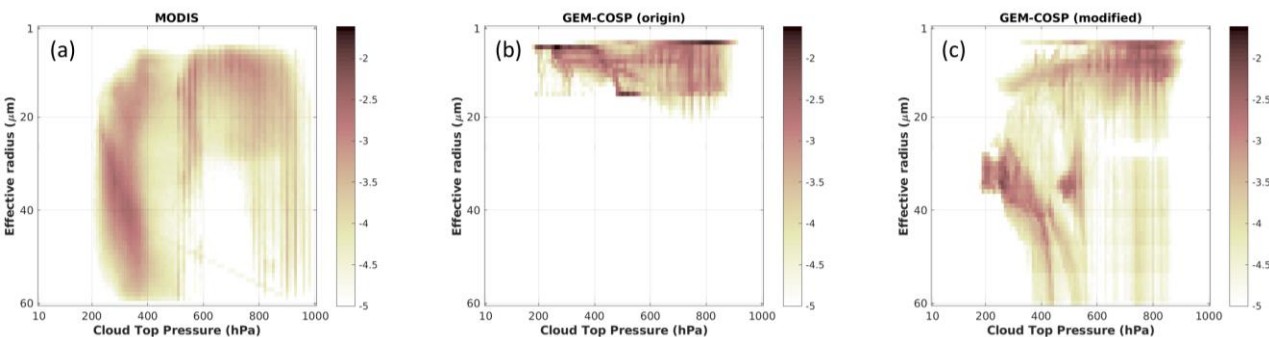

**Figure 19:** Histograms showing numbers of cases in logarithmic scale as functions of effective radius (bin width of 1
μm) and cloudtop pressure (bin width of 10 hPa). (a) MODIS retrievals from MYD06_L2 product. (b) original GEM data simulated by MODIS simulator of COSP. (c) as in (b) but this is for adjusted effective radii.

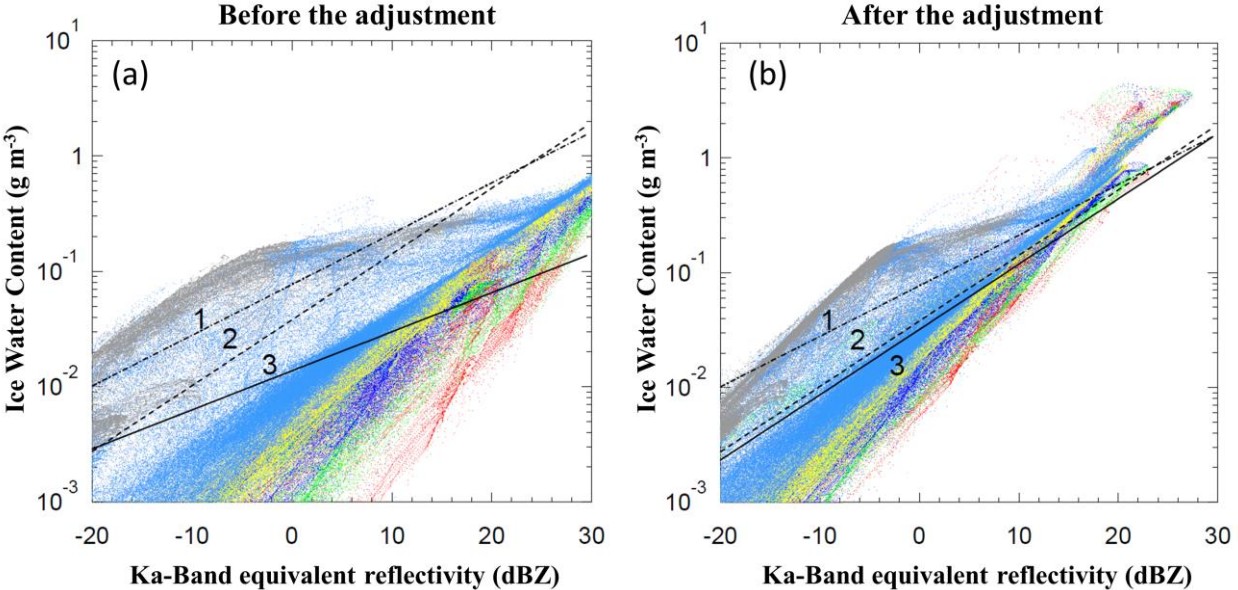

**Figure 20:** IWC vs. Ka-band equivalent reflectivity for the nadir track of the Halifax scene. Lines 1 and 2 are best-fit lines to GCPEX and CRYSTAL-FACE data, respectively, (see Matrosov and Heymsfield 2017). Line 3 is best-fit to
GEM results. Red points are for temperatures $T$ (in $^{\circ}$ C) between 0 $^{\circ}$ and -5$^{\circ}$, Green:-10 $^{\circ}$ < $T$ < -5 $^{\circ}$, Dark-Blue:-15 $^{\circ}$ < $T$ < -10 $^{\circ}$, Yellow: -20 $^{\circ}$ < $T$ < -15 $^{\circ}$, Light-Blue: -40 $^{\circ}$ < $T$ < -20 $^{\circ}$ , Grey: $T$ < -40 $^{\circ}$. (a) is for before adjustment of effective radii, and (b) is for after adjustment.

## 8. Conclusions, perspectives, and data availability

In this day and age, it is difficult to see how a scientifically and technically advanced research satellite could be launched without first having completed a pre-launch *end-to-end* numerical simulation programme that assesses myriad aspects of mission performance and demonstrates the likelihood of achieving the mission's science goals. Such a programme would begin with simulation of atmosphere-surface conditions that ideally span, and resemble, much of what can be expected to be encountered during the mission. Virtual observations, to be made by the satellite's sensors, are then simulated for the mock atmosphere-surfaces, and they are, in turn, operated on by retrieval algorithms. This exact end-to-end simulation programme has unfolded, over the past two decades, for the Earth-CARE satellite mission (ESA 2001; Illingworth et al. 2015). The purpose of this paper was to summarize the atmosphere-surface test datasets.

The synthetic atmosphere-surface systems used for this study were produced by ECCC's Global Environment Multiscale model, which is abbreviated as "GEM" (Côté et al., 1998, Girard et al., 2014). This operational numerical weather prediction model is well-known internationally (Leroyer et al., 2014, 2022; Bélair et al., 2016; Milbrandt et al., 2016; Qu et al., 2018, 2020, 2022a; McTaggart-Cowan et al. 2019). The end-to-end programme was intended initially to test retrieval algorithm performance, but was expanded to address ESA's communication and data handling segments, too. As such, large simulated domains were required. The fundamental data processing element is referred to as a *frame*. There are eight frames per orbit, and so simulated "test frames" had to be ~6,200 km along-track. The across-track swath of EarthCARE's multi-spectral imager (MSI) is 150 km, and so test frames had to be at least this wide, but to avoid edge-effects, their widths were extended to 200 km. Horizontal and vertical resolutions for GEM's simulations had to allow for at least some variability within the foot-prints of EarthCARE's sensors. Use of 57 vertical layers and horizontal grid-spacing $\Delta x$ of 0.25 km were deemed adequate (see Qu et al. 2018).

Three test frames, that followed orbits provided by ESA, were identified via examination of satellite data and available surface weather observations. Conditions that were captured include: a cold frontal system, broken shallow cumulus, a tropical mesoscale convective system, thin cirrus, multi-layer clouds; clear-sky conditions, with aerosols, over ocean, land (including mountains), and ice/snow surfaces. Surface bidirectional reflection distribution functions (BRDFs) and albedos from climatological data and aerosol properties were added to GEM's simulations. GEM's "computational equator" was oriented along each EarthCARE orbit and a set of nested simulations were performed

that culminated in 13, separately simulated, innermost domains at $\Delta x = 0.25$ km. These domains were concatenated

to form the full 6,200 km test frames.

It was discovered that GEM's descriptions of some ice cloud properties lack the realism needed for adequate simulation of virtual observations and assessment of cloud and aerosol property retrieval algorithms. Hence, modifications were made to the effective size of ice particles based on surface and *in-situ* observations. Most of the important

impacts were to particle sizes near cloudtops.

Previous studies have simulated atmospheric conditions for purposes of satellite algorithm development and evaluation (e.g., MPB Technologies Inc. 2000; Voors et al. 2008; Tao et al. 2009), constraint of cloud microphysical schemes by observations (Matsui et al. 2013, Iguchi et al. 2012a, 2012b, 2014), and assimilation of retrieved aerosol properties into Numerical Weather Prediction (NWP) models (Zeng et al. 2020; Cornut et al. 2023), but the simula-

tions done for this study had to serve several purposes simultaneously, and this put unique demands on them. Most notably their size, for they had to provide sufficient detail to meet several wide-ranging aspects of observation simulation and algorithm assessment, with enough areal extent to evaluate data processing and archiving procedures.

As such, the overarching requirement placed on the time-sensitive production of these test frames was that they be deemed, by myriad mission researchers and managers, "sufficiently" realistic and "necessarily" expansive enough to

provide adequate assessment of the numerous key steps that will be required to produce EarthCARE data. No doubt, this requirement compromised some aspects of both the quality of the simulations and their verification against independent sources of information. Moreover, efforts are being made to improve upon these test data. In particular, the two-moment bulk cloud microphysics scheme Predicted Particle Properties (P3) (Morrison and Milbrandt 2015; Milbrandt and Morrison 2016; Cholette et al. 2019; Qu et al. 2022) is being used.

That said, it is felt roundly that the objectives behind the test frames have been realized and that they have the potential to be used for other observation missions that could include platforms other than satellites; particularly those targeting cloud, aerosol, and radiation interactions. The full dataset is available publicly as summarized in the *Data availability* statement at the end. It includes atmospheric and surface properties produced by GEM data, modified hydrometeor properties, climatological surface optical properties, and the properties of added aerosols (cf. tables in

the Appendix).

**Acknowledgements**

This study was made possible by a series of contracts from the European Space Agency. The authors thank Jason Milbrandt, Sylvie Leroyer, Stéphane Bélair, and Manon Faucher for technical help and discussions.


**Author contributions**

HWB, ZQ, JNSC, MWS and DPD conceptualized the research goals and aims. ZQ performed the GEM simulations. DPD and VH adjusted the original GEM cloud data and added the aerosols properties. ZQ, HWB and DPD drafted the manuscript with the contributions from all co-authors.


**Data availability**

Halifax frame's data are available at Qu et al. 2022b,c. Baja frame's data are available at: Qu et al. 2022d,e. Hawaii frame's data are available at: Qu et al. 2022f,g,h.

**Financial support**

This study is supported by Clouds, Aerosol, Radiation - Development of INtegrated ALgorithms (CARDINAL) for the EarthCARE Mission.

**Appendix: List of test frame variables**

**Table A1:** Variables for the original GEM simulations

| Variables name | Units | Dimension | Notes |
|---|---|---|---|
| water_content_cloud | g m$^{-3}$ | 3 | **Explicit Cloud:** |
| water_content_ice | g m$^{-3}$ | 3 | From MY2 double-moment (MY2) scheme. It's an |
| water_content_rain | g m$^{-3}$ | 3 | explicit scheme. The cloud optical properties could |
| water_content_snow | g m$^{-3}$ | 3 | be calculated for each of the six species. There also |
| water_content_graupel | g m$^{-3}$ | 3 | could be properties from the implicit schemes such |
| water_content_hail | g m$^{-3}$ | 3 | as the Planetary Boundary Layer (PBL) and shallow |
| number_concentration_cloud | # m$^{-3}$ | 3 | convection (SC) scheme (see below). Their optical |
| number_concentration_ice | # m$^{-3}$ | 3 | properties could be calculated in a similar way to |
| number_concentration_rain | # m$^{-3}$ | 3 | those for MY2 species. |

| Variable | Units | Dim | Notes |
|---|---|---|---|
| number_concentration_snow | # m$^{-3}$ | 3 | The final optical properties for use should be the combination of those of the concerned MY2 species and of the implicit clouds. |
| number_concentration_graupel | # m$^{-3}$ | 3 | |
| number_concentration_hail | # m$^{-3}$ | 3 | |
| effective_radius_cloud | m | 3 | $R_{eff}$ calculated based on water content and number concentration from MY2 scheme. |
| effective_radius_ice | m | 3 | |
| effective_radius_rain | m | 3 | |
| effective_radius_snow | m | 3 | |
| effective_radius_graupel | m | 3 | |
| effective_radius_hail | m | 3 | |
| implicit_cloud_solid_water_content | g m$^{-3}$ | 3 | Implicit Cloud: Cloud condensates from implicit schemes (PBL+SC). $R_{eff}$ for solid condensate is assumed to be 15 microns. |
| implicit_cloud_liquid_water_content | g m$^{-3}$ | 3 | |
| implicit_cloud_liquid_effective_radius | m | 3 | |
| BRDF_iso | [0 1] | 2 (8 bands) | Snow-free ground albedo climatology from MCD43GF. Ross-Li model (see Eq. 1 and 2). Add information of snow (X-MET, snow_depth etc.). |
| BRDF_vol | [0 1] | 2 (8 bands) | |
| BRDF_geo | [0 1] | 2 (8 bands) | |
| height_thermodynamic | m | 3 | Levels for all the 3D variables except those concerning the horizontal wind. |
| pressure_thermodynamic | Pa | 3 | |
| height_momentum | m | 3 | Levels for the variables concerning the horizontal wind (speed and direction). |
| pressure_momentum | Pa | 3 | |
| temperature | K | 3 | |
| specific_humidity | g m$^{-3}$ | 3 | |
| relative_humidity | [0 1] | 3 | |
| wind_horizontal_speed | m s$^{-1}$ | 3 | On thermodynamic levels |
| wind_horizontal_direction | deg | 3 | 0: north, clockwise, on thermodynamic levels |
| wind_vertical_speed | m s$^{-1}$ | 3 | |
| cloud_mask_3d | 0/1 | 3 | |
| cloud_mask_2d | 0/1 | 2 | |
| orography | m | 2 | |
| solar_zenithal_angle | deg | 2 | |
| surface_pressure | Pa | 2 | |
| water_land_fraction | [0 1] | 2 | 1: 100% lake/sea/ocean, 0: 100% land |
| surface_temperature | K | 2 | Sea ice not included |
| ice_fraction | [0 1] | 2 | Includes sea ice + land ice. To distinguish, use *water_land_fraction*. Use *ice_fraction* to apply ice_temperature. |
| ice_temperature | K | 2 | |
| total _water_path | g m$^{-2}$ | 2 | Liquid cloud water path + ice cloud water path |
| ice_water_path | g m$^{-2}$ | 2 | Ice cloud water path |
| vertical_integrated_water_vapeur | g m$^{-2}$ | 2 | |
| snow_depth | m | 2 | |
| total_precipitation_rate | m s$^{-1}$ | 2 | |
| liquid_precipitation_rate | m s$^{-1}$ | 2 | |
| longitude | deg | 2 | |
| latitude | deg | 2 | |

**Table A2:** Variables archived for the test frames

| Variables name | | Units | Dimension | Notes |
|---|---|---|---|---|
| BRDF_iso | | [0 1] | 2 (8 bands) | Snow-free ground albedo climatology from MCD43GF. Ross-Li model (see Eqs. 1 and 2). NB. Add information of snow (X-MET, snow_depth etc.). |
| BRDF_vol | | [0 1] | 2 (8 bands) | |
| BRDF_geo | | [0 1] | 2 (8 bands) | |
| ice_fraction | | [0 1] | 2 | Includes sea ice + land ice. To distinguish, use *water_land_fraction*. Use *ice_fraction* to |
| ice_temperature | | K | 2 | |

| | | | | apply *ice_temperature*. |
|---|---|---|---|---|
| total_precipitation_rate | | m s$^{-1}$ | 2 | |
| liquid_precipitation_rate | | m s$^{-1}$ | 2 | |
| orography | | m | 2 | |
| snow_depth | | m | 2 | |
| solar_zenithal_angle | | deg | 2 | |
| surface_pressure | | Pa | 2 | |
| water_land_fraction | | [0 1] | 2 | |
| surface_temperature | | K | 2 | |
| surface_wind_speed | | m s$^{-1}$ | 2 | |
| temperature | | K | 3 | |
| specific_humidity | | g m$^{-3}$ | 3 | |
| pressure | | Pa | 3 | |
| wind_horizontal_speed | | m s$^{-1}$ | 3 | |
| wind_vertical_speed | | m s$^{-1}$ | 3 | |
| mass_content | | g m$^{-3}$ | 3 | For 6 types of hydrometeors: liquid cloud, ice cloud, rain, snow, graupel and hail, and for 4 types of aerosols: coarse dust, coarse salt, fine mode weakly absorbing and fine mode strongly absorbing. |
| effective_radius | | micron | 3 | |
| number_concentration | | cm$^{-3}$ | 3 | |

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
