# Peer review of "Numerical Model Generation of Test Frames for Pre-launch Studies of EarthCARE's Retrieval Algorithms and Data Management System"

_Atmospheric Measurement Techniques, 2022_

## Author Comment (AC1)

Dear Editor,

Please find there our responses to the reviewers' comments. Our responses are in blue.

Sincerely,

Zhipeng Qu

**Referee #1 (Toshi Matsui)**

**Summary**: This manuscript describes generation and validation of numerical atmosphere-surface test frames for upcoming EarthCARE satellites. Authors conducted multiple multi-scale modeling to generate high-resolution scenes of virtual atmosphere and surface across different regions. The manuscript, then, qualitatively validated generated virtual fields against the satellite observations. This is a very ambitious paper that intends to validate this many different scenes all together in a single manuscript. This is why the validation method lacks detailed quantitative analysis and non-evident arguments in ice microphysics. Other problems include lack of citations related to previous studies and lack of aerosol validation. Thus, this paper needs "major revision" to resolve aforementioned problems before publishing in AMT. Please read the major and minor comments below.

Thank you for your review and suggestions. We agree that more quantitative verification of the test frames would improve the quality of the manuscript, and some have been included in the revised version, but we also stress that this was not, at this point, the purpose of the manuscript. This is elaborated in our responses to subsequent points.

**Major Comments**:

1) Lack of citations and discussion in the previous studies.

This numerical satellite mission frame has emerged from the last decades. We have developed a benchmark of numerical test frames before the launch of Global Precipitation Measurements (GPM) Core satellite.

Matsui, T. T. Iguchi, X. Li, M. Han, W.-K. Tao, W. Petersen, T. L'Ecuyer, R. Meneghini, W. Olson, C. D. Kummerow, A. Y. Hou, M. R. Schwaller, E. F. Stocker, J. Kwiatkowski (2013), GPM satellite simulator over ground validation sites, Bull. Amer. Meteor. Soc., 94, 1653–1660. doi: http://dx.doi.org/10.1175/BAMS-D-12-00160.1

For this, we have validated numerical simulation against the various measurements from the GPM field campaigns.

Iguchi T., T. Matsui, J. J. Shi, W.-K. Tao, A. P. Khain, A. Hou, R. Cifelli, A. Heymsfield, and A. Tokay (2012), Numerical analysis using WRF-SBM for the cloud microphysical structures in the

C3VP field campaign: Impacts of supercooled droplets and resultant riming on snow microphysics, Journal of Geophysical Research, 117, D23206, doi:10.1029/2012JD018101.

Iguchi, T., T. Matsui, A. Tokay, P. Kollias, and W.-K. Tao (2012), Two distinct modes in one-day rainfall event during MC3E field campaign: Analyses of disdrometer observations and WRF-SBM simulation. Geophysical Research Letters, 39, L24805, doi:10.1029/2012GL053329.

Iguchi, T., T. Matsui, W. Tao, A. Khain, V. Phillips, C. Kidd, T. L'Ecuyer, S. Braun, and A. Hou, 2014: WRF-SBM simulations of melting layer structure in mixed-phase precipitation events observed during LPVEx. J. Appl. Meteor. Climatol. 53, 2710-2731, doi:10.1175/JAMC-D-13-0334.1.

There should be a lot more papers related to these topics. Please search manuscripts and discuss similarities and differences between your work and other previous work in the introduction section.

Thank you for these suggestions. New text and references have been added to the introduction. While we agree with the Reviewer that verification of the test frames using independent sources of information is a worthy and very demanding task, we emphasize, in the revised version, that the test frames, as described in this manuscript, were produced expressly for end-to-end assessment of EarthCARE's algorithms and its data-handling procedures. While the frames had to be "reasonably realistic" and sufficiently large and demanding for retrieval algorithms to be tested rigorously, they were never expected to be "perfect facsimiles" and the manuscript was never intended to focus on verification of the frames via use of independent observations. In fact, given the size of the test frames, extensive verification of them would be extremely challenging and in all likelihood unsatisfactory. We tried to give a cursory indication.

"Using numerical simulation to generate atmospheric conditions for satellite algorithm development and testing is not new. These atmospheric conditions could be idealized with different spatial resolutions from tens of km to tens of meters targeting different types of cloud scene (Park et al. 2000, Voors et al. 2008). Tao et al. (2009) use multi-scale modelling system combining cloud resolving model (CRM), numerical weather prediction (NWP) model and general circulation model (GCM) with unified physics to produce atmospheric scenes at different scales. More detailed size-bin-resolving cloud microphysics scheme could also be used to produce atmospheric conditions which could be subsequently evaluated and constrained by the observations (Matsui et al. 2013, Iguchi et al. 2012a, 2012b, 2014). Zeng et al. (2020) combine high-resolution simulation with data assimilation system to produce higher quality of atmospheric scenes. Cornut et al. (2023) use also data assimilation method to incorporate MODIS retrievals of aerosols properties into the NWP simulations."

"Although different methods exist to enhance the quality of simulated atmospheric scenes, due to the known issues in numerical simulation of atmospheric condition, such as spatial/temporal displacement of simulated clouds, poor understanding of many cloud microphysical processes such as secondary ice production (Korolev and Leisner 2020), bulk assumptions in cloud microphysical schemes, etc., simulated atmospheric conditions will always be different from the reality. The main purpose of an end-to-end simulation with radiative closure assessment is to provide an efficient pathway to evaluate the performance of the retrieval algorithm of clouds and

aerosols without focusing on whether the test data represent exactly the reality. The goal then shifted to verify whether these properties are realistic enough within reasonable range and with good relationship among different variables."

2) Lack of quantitative validation.

This paper intends to validate many different scenes that contain many different types of clouds and microphysics. All validation here is qualitative "eyeballs" validation. Thus, the results are stated "very good (line 209). At least, I suggest creating histograms or PDFs for each plot (Figs. 8, 9, 11, 12, 14) for more quantitative discussion in addition to existing qualitative validation. You can see Fig 1d in Matsui et al. 2014 for example.

Matsui, T., J. Santanello, J. J. Shi, W.-K. Tao, D. Wu, C. Peters-Lidard, E. Kemp, M. Chin, D. Starr, M. Sekiguchi, and F. Aires, (2014): Introducing multisensor satellite radiance-based evaluation for regional Earth System modeling, Journal of Geophysical Research, 119, 8450–8475, doi:10.1002/2013JD021424.

Thank you for this suggestion. We added PDFs to figure 8, 9, 11, 12, 14 and 15. Additional discussions are also added in the main text.

[Figure]

**Figure 8:** (a) MODIS TOA flux (approximated by radiance*π) for band 3 (459 - 479 nm) between 17h15 and 17h35 UTC on 7-Dec-2014. (b) RRTMG simulated upward TOA flux for 441.5 - 625 nm for GEM's simulation of the *Halifax frame*. (c) as in (a) but for band 31 (10.8-11.3 μm). (d) in as but for wavelengths 10.2 - 12.2 μm. Solid and dashed yellow lines indicate EarthCARE's and CloudSat's nadir-tracks. Blank areas are outside MODIS's field-of-view. (e) fraction of case for visible bands with regard to spectral fluxes (bin size of 10 W m$^{-2}$ μm$^{-1}$ is used). (f) fraction of case for infrared bands with regard to spectral fluxes (bin size of 0.2 W m$^{-2}$ μm$^{-1}$ is used).

[Figure]

**Figure 9:** (a) MODIS cloudtop pressure (MYD06_L2 product) between 17h15 and 17h35 UTC on 7-Dec-2014. Blank areas are outside MODIS's field-of-view. (b) GEM's cloudtop pressure for the *Halifax frame* based on COSP's MODIS simulator. Grey area in the northern portion has $\theta_0 > 90°$ and so no COSP values. (c) fraction of case with regard to cloudtop pressure (bin size of 10 hPa is used).

"Figure 8a and c show MODIS spectral fluxes (MYD02HKM product; MCST 2017a) for 0.459 - 0.479 μm and 10.8 - 11.3 μm for the Halifax frame. Key cloud-related features are a cold front between 40°N and 45°N, scattered clouds to its south, and mostly overcast conditions to its north. Figure 8b and d show TOA spectral fluxes for two wavebands, close to MODIS's bands, as simulated by the Rapid Radiative Transfer Model for GCMs (RRTMG - Mlawer et al. 1997; Iacono et al. 2000; 2008) using GEM data. At large-scales, GEM did well with respect to cloud occurrence. Figure 8e and f show the fraction of cases of visible and infrared bands with regard to different spectral fluxes, respectively. The frequencies of different spectral fluxes for both visible and infrared wavelength follow similar patterns, although some differences remain. For the GEM scenes, discontinuities, stemming from the stitching together of the semi-independent high-resolution inner-most domains, are clearly visible across the frontal system. They do not pose a serious problem for the task at hand."

"Near 38°N GEM's longwave fluxes are significantly less than MODIS's. This is because GEM simulated widespread convection in this area whereas MODIS only observed isolated convective cells. This can be seen in Figure 8e that GEM-RRTMG shows higher frequency around 800 W m$^{-2}$ μm$^{-1}$, and in Figure 8f that a higher frequency is observed from GEM-RRTMG around 5 W m$^{-2}$ μm$^{-1}$. This is also apparent in Figure 9a and b, which shows cloudtop altitudes both inferred from MODIS radiances (Platnick et al. 2015) and computed by the MODIS simulator of the Cloud Feedback Model Intercomparison Project (CFMIP) Observation Simulator Package (Bodas-Salcedo et al. 2011; abbreviated as the COSP simulator). This is consistent with the overestimation of the simulated frequency by GEM-COSP (Figure 9c) near 200 hPa."

"GEM also simulated slightly higher cloud top height for low-level clouds for the southern section between latitude 20 and 30°N and the northern section between latitude 50 and 55°N. The frequency of GEM simulated low-level clouds in Figure 9c is centered on 750 hPa, whereas MODIS retrieved clouds are lower, mostly centered on 920 hPa. This can also be inferred from Figure 8f in which higher frequencies of infrared spectral fluxes from GEM-RRTMG are found between 23 and 25.5 W m$^{-2}$ μm$^{-1}$ for the southern section and between 10 and 12 W m$^{-2}$ μm$^{-1}$ for the northern section. In addition, in Figure 9c, we also find that GEM underestimated the amount of mid-level clouds between 500 and 600 hPas. This corresponds to the region between the latitude 55 and 62°N."

"Figure 11 compares MODIS TOA fluxes to those computed by RRTMG acting on GEM data for the Baja frame. As with the Halifax frame, agreement is generally good, though GEM's fields exhibit some peculiarities. For instance, GEM's fluxes associated with clouds are less variable than MODIS's; especially between 40°N and 50°N. This could be due to both GEM's clouds being simply too homogeneous due to missing mesoscale forcing or RRTMG's use of 1D radiative transfer models (see Barker et al. 2017). Also, the thin high clouds near latitude 32°N, which are also evident in Figure 12 and positioned well in space, show an on-off pattern that is not seen in the observations. Furthermore, near latitude 55°N GEM failed to produce the very low and thin, but extensive, clouds. This is most apparent in Figure 12a and b. Figure 12c shows clearly the missing of these low-level clouds below 800 hPa. Finally, GEM simulated larger amount of cloud with cloudtop pressure of ~400 hPa near latitude 50°N. This is consistent with the frequency plot (Figure 12c) in which an overestimation by GEM at ~400 hPa can be clearly seen. This can also be confirmed from Figure 11e and f in which significant overestimation of frequency can be found between 620 and 730 W m-2 μm-1 for visible band and between 7 and 10 W m$^{-2}$ μm$^{-1}$ for infrared band. Note too, that the discontinuities that stem from stitching together GEM's innermost domains are less apparent for this frame than they are for the Halifax frame, though the discontinuity near 26°N is notably bad for it stands out in both visible and IR imagery."

[Figure]

**Figure 11:** As in *Figure 8* except these are for the *Baja frame*. MODIS TOA fluxes are observed between 20h10 and 20h30 UTC on 2-Apr-2015

[Figure]

**Figure 12:** As in *Figure 9* except these are for the *Baja frame*. MODIS cloudtop pressure are retrieved between 20h10 and 20h30 UTC on 2-Apr-2015.

"Figure 14 shows that for the Hawaii frame, GEM's positionings and approximate intensities of cloud systems near the Equator and ~25°S agree well with the MODIS observations. The harsh discontinuity in GEM's string of innermost domains near 2°S is due to a lack of high ice cloud, as seen in Figure 15, which likely stems from the lack of information, in the form of reduced outflow of high cirrus, coming into the sub-domain from the equatorial mesoscale system. Likewise, near 15°N the lack of upper-level cloud in GEM could be because this sub-domain was too disconnected from the mesoscale system to the south. The missing of high clouds in the simulation can inferred from Figure 14e with an overestimation of frequency by GEM near 200 W m$^{-2}$ μm$^{-1}$ which resembles TOA visible fluxes from ocean surface. This missing of high clouds can also be confirmed by Figure 14f with higher frequency by GEM for warmer surface near 27 W m$^{-2}$ μm$^{-1}$. This can also be inferred from Figure 15c in which underestimation of frequency by GEM between 100 and 200 hPa is found. Again, however, the point of this section is to show the gross verisimilitude of the test frames and hence their suitability for EarthCARE algorithm assessments."

[Figure]

**Figure 14:** As in *Figure 8* except these are for the *Hawaii frame*. MODIS TOA fluxes are observed between 00h35 and 00h55 UTC on 24-Jun-2015

[Figure]

**Figure 15:** As in *Figure 9* except these are for the *Hawaii frame*. MODIS cloudtop pressure is retrieved between 00h35 and 00h55 UTC on 24-Jun-2015

I also suggest utilizing more detailed satellite simulators to generate observation-equivalent scenes. For example, RRTMG is broad-band RT. The COSP simulator does not account for details in size and phase of MY2. Detailed satellite simulator must closely follow assumptions in size distributions, phase, and shapes in cloud microphysics. You can read section 2 of above paper for more principles and radiance-based model evaluation.

The results presented in this manuscript are meant to provide an overview of the ability of the simulations to provide "reasonably realistic and large" test frames. Given this goal, we used output from the COSP simulator and RRTMG radiative fluxes. While they are not as complex or comprehensive as other systems used to simulate observations, they are one step toward the use of similar quantities to compare model output and observations. Other manuscripts in this special issue present results of applying very sophisticated instrument simulators (ATLID, CPR, MSI and BBR) to the test frames (the list of the manuscripts).

3) Non-evident argument of ice microphysics bias and improvement.

Section 7 argument suddenly starts with "Basically, GEM predicts too many overly small ice crystals….". Unfortunately, I don't see any such evidence in this manuscript or previous manuscript using MY2 microphysics. What is this argument based upon? You must show evidence using either observations or previous manuscript using MY2 in different cloud types. Then, discussion goes to "these alterations were found to …. Improvement of GEMS's simulated cloud properties…." Again, based on what?? No evidence. Essentially Figure 17 compares Reff-T distributions before and after modification.

There are potential pathways to provide this evidence.

1. Validate against Reff products of MODIS/VIIRS satellites. Although these products have their own assumptions, it is better than nothing.
2. Simulate CloudSat reflectivity before and after the change in ice size. In this case, you must use a detailed radar simulator that accounts for size, phase, and non-sphericity of ice crystals.

Reff is the inverse function of lambda in generalized gamma distributions, thus, change in Reff can significantly impact CloudSat radar reflectivity. You can construct contoured frequency of

altitude diagrams (CFADs) or similar statistical composites. See examples Fig 10 & 11 of Shi et al. 2010 for example.

Shi, J. J., W.-K. Tao, T. Matsui, A. Hou, S. Lang, C. Peters-Lidard, G. Jackson, R. Cifelli, S. Rutledge, and W. Petersen (2010), Microphysical Properties of the January 20-22 2007 Snow Events over Canada: Comparison with in-situ and Satellite Observations. Journal of Applied Meteorology and Climatology. 49(11), 2246–2266.

Thank you for the suggestions. Assessment of adjustments to Reff and resulting improvements were addto the manuscript. The focus of the assessment is whether the adjustments improved the relationship between Reff and other variables (e.g., Reff_i vs. T; Reff_i vs. cloudtop P; and IWC vs. Reflectivity) instead of assessing whether the simulated Reff values agree well with observations.

"The impact of the adjustment of effective radius can be seen in 2D histograms in Figure 19. Figure 19a shows the fraction of cases from MODIS retrievals (MYD06_L2) with regard to effective radius and cloudtop pressure. The effective radii of most cases with cloudtop pressure between 200 and 400 hPa are between 30 and 50 μm. For the COSP simulated results based on the original GEM data, most of the ice clouds with same range of cloudtop pressure have very small effective radius, mostly smaller than 15 μm (Figure 19b). After the adjustment, the results improved significantly with most cases having effective radius between 30 and 50 μm which agrees well with the MODIS retrievals (Figure 19a)."

[Figure]

**Figure 19:** fraction of cases in logarithmic scale. Y-axis: effective radius with bin size of 1 μm. X-axis: cloudtop pressure with bin size of 10 hPa. (a) MODIS retrieval from MYD06_L2. (b) original GEM data simulated by MODIS simulator of COSP. (c) GEM data after the adjustments of effective radius simulated by COSP.

"In addition to these improvements in cloud optical properties, the same adjustments were found to improve the realism of cloud properties that are relevant to simulation of CPR observations. For example, after applying the adjustments the relationship between ice CWC and simulated radar reflectivity now falls in phase-space that agrees well with real observations (e.g., Matrosov and Heymsfield 2017; Heymsfield et al. 2005). Figure 20 shows the IWC-vs-Ka Band reflectivity for the nadir Halifax scene path. It can be seen that after the adjustment (Figure 20b) the best-fit line of GEM data compares well with the relationships shown in Fig 4 of Matrosov and Heymsfield et al. (2017). The agreement is even more striking if the distribution of the data shown in Matrosov and Heymsfield et al. (2017) is considered instead of just the best-fit lines (not shown). Various

cross-sections of adjusted GEM+CAMS-derived fields can be found in the Supplementary Material."

[Figure]

Figure 20: IWC vs Ka band equivalent reflectivity for the nadir track of the Halifax scene. Lines 1 and 2 are the best-fits line to GCPEX and CRYSTAL-FACE respectively data taken from Matrosov and Heymsfield et al. (2017). Line 3 is the best-fit data to the GEM results. The Red points are for temperatures between 0 and -5 C, Green:-10<T<-5 C, Dark-Blue:-15C<T-10C, Yellow: -20<T<-10C, Light-Blue: -20C<T-40C, Grey: T<-40C. (a) the relationship before the adjustment. (b) the relationship after the adjustment.

4) Lack of aerosol validation.

Although aerosols are part of the numerical testing frame, only cloud properties are validated. Aerosols are supposed to be one of the major components of EarthCARE satellites, right? Why not validate aerosols? I understand that it is incorporated from the CAMS field, and applied ECSIM scattering properties. You can simulate these quantities and present statistical quantities. Are these realistic against existing Lidar measurements, like CALIOP/CATS sensors? See example of how to validate aerosol total backscattering in Figs 16 of Choi et al. 2020.

Choi, Y., S.-H. Chen, C.-C. Huang, K. Earl, C.-Y. Chen, C. Schwartz, and T. Matsui (2020), Evaluating the impact of assimilating aerosol optical depth observations on dust forecasts over North Africa and the East Atlantic using different data assimilation methods, Journal of Advances in Modeling Earth Systems, 12, e2019MS001890. https://doi.org/10.1029/2019MS001890

Validation of aerosols is beyond the scope of this manuscript. As mentioned in the Supplementary Materials, our goal was limited to producing 'realistic enough' aerosol fields for the purposes of algorithm development and testing. Hence, a large degree of tolerance for ad-hoc choices and procedures was approved and accepted by all involved. Regarding general validation of CAMS's aerosols properties, we added text and references to relevant CAMS papers (e.g., Flemming et al. 2017).

Flemming, J., Benedetti, A., Inness, A., Engelen, R. J., Jones, L., Huijnen, V., Remy, S., Parrington, M., Suttie, M., Bozzo, A., Peuch, V.-H., Akritidis, D., and Katragkou, E., 2017: The CAMS interim Reanalysis of Carbon Monoxide, Ozone and Aerosol for 2003–2015, Atmos. Chem. Phys., 17, 1945–1983, https://doi.org/10.5194/acp-17-1945-2017.

**Minor Comments:**

Title: Title including "Data Management", but I don't see particular discussion in DM. Either omit DM from title, or add more plentiful discussion of DM.

We prefer to keep "Data Management" in the title. We added the following description at the end of the second last paragraph of the Introduction:

"Use of high-resolution, full-frame datasets in ECSIM not only allows for assessment of the quality of EarthCARE's geophysical retrievals, it also facilitates preparation of operational computational resources and management of data produces to be generated during the mission."

Line 13: Write acronym NWP in abstract.

We changed the text to "Numerical Weather Prediction (NWP)"

Line 39: "ensemble" does not sound right. It probably means "diverse surface-atmosphere scene"

We changed the wording to "surface-atmosphere conditions"

Line 112-115: Is PBL scheme and shallow convection scheme applied to which domains (coarse grid only)?

We added a statement at the end of the paragraph:

"Both schemes are used in all domains."

Elsewhere: "inner-domains" should be "inner-most domains".

The changes were made.

Line 127: "Saved variables are listed in the Appendix" but I guess it's also available in Table 5??

We agree this could be confusing, so we've renamed them as Table A1 and Table A2.

Line 138-142: Equations of spectral and white albedo can be omitted since they're not very important for this paper. Just a reference is enough or say "white albedo is used for diffuse radiation".

For the convenience of readers, we prefer to keep the equations in the manuscript so that users of the test frames can compute the albedos.

Line 149: Please put a citation for "another emissivity database", although I understood it is more discussed in the companion paper.

The reference is added into the manuscript.

"Huang, X., Chen, X., Zhou, D. K., & Liu, X.: An observationally based global band-by-band surface emissivity dataset for climate and weather simulations. J. Atmos. Sci., 73(9), 3541– 3555. https://doi.org/10.1175/jas-d-15-0355.1, 2016."

Line 172: Omit sentence "While annoying to look at,"

The change was made.

Line 187 & Figure 10: "making a comparison to GEM useless" This statement is too strong.

We changed the wording to be,

"Unfortunately, CloudSat's retrieval of liquid CWC is problematic (e.g., Li et al. 2018). The comparison of liquid CWC is therefore not shown."

Line 195: 1D RT is not the major reason to create homogeneous brightness temperature. Mis-representation of cloud structures due to missing mesoscale forcing should be the main reason.

We change the phrase as:

"This could be due to GEM's clouds being simply too homogeneous due to missing mesoscale forcing as well as RRTMG's use of 1D radiative transfer models (see Barker et al. 2017)."

Line 209: "very good" is too qualitative and vague a statement (not scientific statement).

We changed the phrase as:

"Figure 14 shows that for the Hawaii frame, the location and intensity of cloud systems simulated by GEM near the Equator and ~25°S  agrees well with MODIS observations."

Line 230: "by algorithm development groups (….." should be "in the previous section."

We changed the phrase as:

"While the macrophysical cloud properties simulated by GEM were deemed satisfactory in the previous section, it was clear that there were shortcomings with its predicted ice cloud microphysical properties (cf. Qu et al. 2018)."

---

## Author Comment (AC2)

Dear Editor,

Please find there our responses to the reviewers' comments. Our responses are in blue.

Sincerely,

Zhipeng Qu

**Anonymous Referee #2**

**Summary**: The paper describes the creation of the test frames for testing the EarthCare retrieval algorithms. As such it provides substantial utility, which is justification for publication in principle. Publication in practice is to be decided based on its ability to fulfill its objective, which I judged based on the clarity and completeness of the description. Overall it passes with flying colors, as it provides a clear, concise, and compelling description of what is done and what is available. The authors are to be congratulated.

Thank you very much for your review and compliment! Please find below our response to each point.

I only have minor editorial comments or suggestions that the authors may want to consider for their final revision.

1.  Line 26: Perhaps say "early to mid 2024 or perhaps later".

    Thank you for the suggestion. We change the phase to:

    "which is scheduled for launch in early- to mid-2024"

2.  Line 42: "Lacks this luxury" is a rather conversational way to make the point which might take readers a few passes to digest.

    Agreed. We changed the phrase as:

    "One could stop here and assess performance by comparing retrieved geophysical quantities to their simulated counterparts (cf. Mason et al. 2023), but obviously in the real mission it is impossible to conduct routine comparisons between retrievals and what is actually present."

3.  Line 63: I was a bit puzzled by the reference to the bin-resolved, as what a bin scheme can resolve is a non-parameteric distribution. Passing a parametric distribution to a bin scheme leads to a lack of resolution and seems simply a matter of practicality when interacting with the radiation, as such this strikes me as an unnecessary detail, elaboration, that is not necessary to understand the present paper.

    Agreed. We made the following change:

"Bulk properties of atmospheric attenuators, such as 3D distributions of GEM's cloud water contents (CWC), are used in conjunction with assumed aerosol/cloud size distributions in order for ECSIM to produce physically-consistent synthetic measurements for each of EarthCARE's sensors."

4. line 76: I don't think I fully understood the rationale for not considering night scenes. The simplicity assumption would be that nocturnal situations don't fundamentally sample a different meteorology, which might be true, but it should be stated, rather than simply focusing on the effect on the instruments.

Agreed. We changed the phrase as:

"With a simply assumption that night-time atmospheric conditions are not fundamentally different from day-time conditions, night retrievals can be approximated by neglecting MSI solar channels and solar back-ground for ATLID."

5. Fig : I would have preferred a qualitative coloring of the frames, and a label of the colors

We made changes for Figure 1:

[Figure]

Figure 1: Examples of several successively numbered EarthCARE orbits as provided by ESA. Frames are colour-coded. The test frames are indicated by shaded areas. Frames 39316D, 39318D, and 39320E are referred to as "Halifax", "Baja", and "Hawaii", respectively.

6. line 84: Why not use the ISO-8601 standard for date formatting.

   Thank you for this suggestion. The format of the dates is changed to ISO-8601 standard.

7. line 103: I know the phrase non-hydrostatic primitive equations is used, but I find it confusing because I think of the hydrostatic assumption one of the things that make the primitive equations the primitive equations. I would prefer, the non-hydrostatic extension of the primitive equations.

   Agreed. The change was made in the manuscript.

8. Fig 5: For domains 2 and 3, I inferred that they are implemented 13 times, for each of the instances of domain 4, but if this could be said more explicitly it would avoid confusion arising from Fig. 5 which shows just one instance.

   Thanks for this suggestion. The phrase is now changed to:

   "The downscaling transitional domains at Δx of 2.5 km and 1 km adapt themselves to the locations of the Δx=0.25 km domains (both domains at Δx of 2.5 km and 1 km are repeated 13 times). A common Δx=10 km domain was used for all 13 segments."

9. Fig 6 - wouldn't a binary color scale be more appropriate for what I infer to be a binary mask.

   The "water-land mask" and "ice fraction" variable are actually continuous values between 0 and 1 (e.g. 50% of grid is land), although most of cases are either 0 or 1. Given this we prefer to keep the continued color map. However, the variable name "water-land mask" is confusing and we have changed it to "water-land fraction". We also changed the description in the caption.

[Figure]

Figure 6: All panels are for the Baja frame (see Figure 1 and Figure 3) and each panel's title is self explanatory. For (b) and (c), blue (fraction of 0) corresponds to 100% land and yellow (fraction of 1) to either 100 % water or 100% ice.

10. line 134: April is not that late in spring, so I was surprised by how little snow there was in the Rockies, making me wonder if this was a bias, or just a false expectation on my part.

From Figure 6d, we can still find some areas with snow depth of ~30 cm near 44°N. The snow information is based on NWP model outputs using a global surface analysis (relatively low resolution), hence there might be inaccuracies. However, since the primary purpose of the test frames is for end-to-end simulation, we, and other algorithm developers, considered inaccuracies and uncertainties such as these to be acceptable.

11. line 203: I thought the 'quite good' was a bit of an overstatement. I guess it depends on one's expectations, and raises the question as to whether the qualitative judgments that are made in these sections are appropriate.

We replaced the phrase with:

"Despite these discrepancies, Figure 13 shows that in the vicinity of where the satellite tracks intersect, vertical realizations of clouds from both GEM simulations and CloudSat retrievals indicate smooth mid-level low density clouds, although those clouds from GEM is more extensive. The altitudes of GEM's clouds over the Rooky Mountains are also in fair agreement with CloudSat's. Unlike the *Halifax frame*, the magnitudes of modelled and "observed" IWPs agree quite nicely, in general."

With regard to the qualitative judgement in the manuscript, we added PDF plots in Figs. 8, 9, 11, 12, 14. More discussions are also added in the revised manuscript. Please refer to the answers to reviewer #1 for more details.

---

## Author Response (AR2)

This paper presents the generation and verification of the numerical situation frames for the EarthCARE L1 data. The simulations have been done with higher resolution than the footprints of the instruments of EarthCARE except the ATLID. And the results are reasonable compared to the observational data. However, I am confused by some sections. I think the authors need to explain the online simulation, offline simulation or just addition of data for the EarthCARE L1 data in section 4, 5 and section 7.

Thanks for the review and the valuable comments. Please find below our answers to your questions (in blue).

**Major comments**

1. Lack of descriptions of the NWP model setup in Section 3

- For realistic simulation, I would like to know the initial condition and lateral boundary information. And what kind of nesting method is used in this study?

Thanks for this remark which will be useful for the readers. We added the following description in the manuscript:

"The global analysis data used in Environment and Climate Change Canada's Global Deterministic Prediction System (GDPS) (Buehner et al. 2015) were used as the initial conditions for the outermost simulation domain at 10 km grid-spacing. The GDPS predictions are used as the lateral boundary conditions with the nesting method described in Thomas et al. (1998)."

References:

Buehner, M., McTaggart-Cowan, R., Beaulne, A., Charette, C., Garand, L., Heillette, S., Lapalme, E., Laroche, S., Macpherson, S. R., Morneau, J., and Zadra, A., 2015: Implementation of deterministic weather forecast systems based on ensemble-variational data assimilation at Environment Canada. Part I: The global system. Monthly Weather Review, 143, 2532– 2559, DOI: https://doi.org/10.1175/MWR-D-14-00354.1.

Thomas, S. J., Girard, C., Benoit, R., Desgagné, M., and Pellerin, P., 1998: A new adiabatic kernel for the MC2 model, Atmosphere-Ocean, 36:3, 241-270, DOI: 10.1080/07055900.1998.9649613.

- Do you overlap the separated domains at 250m resolution?

No. The innermost domains at 250 m are not overlapping with each other. To clarify this point, we change the phrase in section 3 as:

"It was simplest to align GEM's computational equator approximately along EarthCARE's orbit, and divide 6,200 km long frames into 13 **_non-overlapping_** inner-most domains (Δx=0.25 km) and run them separately"

- I am wondering about handling with the cloud fractions for the low clouds. Did you use the shallow convection scheme for the 250m simulations?

Yes. As mentioned in section 3, the shallow convection and boundary layer schemes are used in all simulation domains.

- What kind of turbulence schemes are used? Have you changed the turbulence scheme for 250m simulations?

The turbulent kinetic energy (TKE) scheme is used for 250 m simulation. Yes, there are some modifications of the turbulent parameterization for the 250 m simulation. We added the following description in the manuscript:

"The atmospheric turbulence is parameterized with a turbulent kinetic energy (TKE) scheme (Benoit et al. 1989, Bélair et al. 2005) named MoisTKE. For the simulations with 250 m horizontal grid-spacing, a modified mixing length with an asymptotic value based on the horizontal grid size [$\lambda_0 = 0.23(\Delta x\, \Delta y)^{1/2}$] is used. The readers are refer to Leroyer et al. (2014) for more details."

References:

Benoit, R., J. Côté, and J. Mailhot, 1989: Inclusion of a TKE boundary layer parameterization in the Canadian regional finite-element model. Mon. Wea. Rev., 117, 1726–1750, doi:10.1175/1520-0493(1989)117<1726:IOATBL>2.0.CO;2.

Bélair, S., J. Mailhot, C. Girard, and P. Vaillancourt, 2005: Boundary layer and shallow cumulus clouds in a medium-range forecast of a large-scale weather system. Mon. Wea. Rev., 133, 1938–1960, doi:10.1175/MWR2958.1.

Leroyer, S., Bélair, S., Husain, S., and Mailhot, J., 2014: Subkilometer numerical weather prediction in an urban coastal area: a case study over the Vancouver metropolitan area, J. Appl. Meteorol. Clim., 53, 1433–1453, https://doi.org/10.1175/JAMC-D-13-0202.

2.Section 4

I understand the shortwave optical properties of the numerical model is not good. I am confused with these data are used in the simulations in the radiation scheme to influence the clouds and precipitation or just added data after the simulation.

The shortwave optical properties of land surface described in the manuscript is not used in the NWP simulations. They are only used as added data for the pre-launch studies of EarthCARE. We modified the manuscript as follow:

"As the additional data for pre-launch studies of EarthCARE, GEM's snow-free surface albedos were replaced by those based on MODIS's MCD43GF 1 km resolution bidirectional reflectance distribution function (BRDF) product for the period 2002 to 2013 (Schaaf et al. 2002)."

3. Section 5

-Describe more detailed information about the data for CAM, such as horizontal resolution and vertical resolution.

- What types of aerosol optical properties are used in this study?

The text around at the beginning of section 5 has now been changed to:

"The ECSIM scene creation process requires 3D distributions of aerosol size distributions. As GEM lacks interactive aerosol tracers and chemistry, aerosol fields were added to the test scenes using information from the Copernicus Atmosphere Monitoring Service (CAMS) (Flemming et al. 2017). The CAMS data was at a resolution of 0.5 by 0.5 Lat-Lon degrees and 60 hybrid sigma model levels. The aerosol scheme implemented within ECSIM follows the *Hybrid End-To-End Aerosol Classification* (HETEAC) approach of defining a certain set of basic aerosol types with associated e.g. size distributions, refractive indices and optical properties that, when weighted and summed, yield adequate representations of a wide range of observed aerosol optical properties (Wandinger et al. 2016; Wandinger et al. 2023). Table 4 lists the CAMS aerosol fields, and the *Supplementary Material* section provides a detailed description of the mapping between CAMS fields and ECSIM/HETEAC scattering types. It also provides more details regarding aerosol representation."

-Have you used these data in your simulation (off-line simulation)?

If so, could you explain how to interact with radiation or microphysics shorty?

The added aerosols are not used in NWP (on-line or off-line) simulations.

4. Section 7

- Have you applied these changes to the previous results?

The results presented before the section 7 didn't include the changes.

Are they only for the EarthCARE L1 data or are they just an introduction to an update plan?

The modifications are applied to the original GEM data. These modified data then are used in the EarthCARE simulations to produce L1 and L2 products.

If you used these modifications for the previous results. Please go to the upper parts after section 3.

As previously answered, the modifications are not included in the results before the section 7.

- How about the effect of the changes on the thin cirrus? I think your simulations reproduced the thin cirrus more than the observation (Fig. 10, 13, 15). However, your improvement increases the thin cirrus in Fig. 18. Please explain this.

The panels (b) of Fig. 10, 13 and 16 show CloudSat's 94 GHz radar retrievals of IWC which are not suitable to detect thin cirrus cloud. This might partly explain the lower amount of thin cirrus cloud from the CloudSat data. However, in Fig. 15, we do see the missing high ice clouds between -2° and -6°S. But it is more likely to be caused by the harsh discontinuity in GEM's string of innermost domains as explained in the subsection 6.3.

The perceived changes the thin cirrus noted by the reviewer are spurious. The figure caption has been updated to make this clear.

"Nadir cross-sections of $R_{eff}$ and 355 nm extinction before and after making adjustments described in the text. Note that the changes in the aerosol regions e.g. the elevated layer north of 50 Degrees at around 7.km are due to technical updates to the aerosol processing between the "Before" and "After" data not related to the ice-cloud adjustments."

**Specific comments**

L47: Six or eight frames per orbit? I think eight frames per orbit.

Thanks for the remark. Indeed, we changed the text accordingly.

L77: please check the number of frames per orbit.

Agreed. We changed the text accordingly.

L77-79: I think the name of frames are depending on the latitude. Please check the criterion of the frames.

Thanks very much for this comment. We modified the description in the section 2.

"Figure 1 shows several EarthCARE orbits, numbered 39316 through 39320. An orbit consists of eight frames; each frame's number having an appending letter from A to H which is defined by given ranges of altitude (JAXA, 2017). Frames are colour-coded and measure ~5,000 km along-track and 150 km across-track."

L85: I think the main reason for the choice of orbit is the availability of A-train data (CloudSat and CALIPSO). The swath of CloudSat and CALIPSO is very narrow compared to the other satellite. Please move it to the upper part.

We considered that both the various surface/atmospheric conditions and the interception with A-Train satellites are equally important. Hence we prefer to keep the statement as is.

L249: There is a difference between observation and simulation of ice precipitation between 42N and 43N in Fig. 10. GEM produced the ice precipitation as snow or graupel. Has GEM underestimated the surface temperature in these areas? I suggest some comments on this.

The near surface areas between 42°N and 43°N are indicated as "Significant return power but likely surface clutter" in CloudSat's CPR_Cloud_mask data (mask=5). This means that the retrievals are not reliable. We excluded these data in the results presented in Fig. 10. Therefore, it doesn't mean that there is not ice

precipitation in the area. Unfortunately, there is not surface data available for the verification. We added a short comment in subsection 6.1:

"Between 42°N and 43°N, GEM produces a large amount of solid precipitation whereas in CloudSat data, due to the ground clutter, there is no reliable retrieval available."

L412: six or eight frames per orbit? I have commented on this part in the comments above.

It should be eight frames. We changed the text.